# Hyperflows:
# Pruning Reveals the Importance of Weights

## Abstract

Network pruning is used to reduce inference latency and power consumption in large neural networks. However, most existing methods struggle to accurately assess the importance of individual weights due to their inherent interrelatedness, leading to poor performance, especially at extreme sparsity levels. We introduce Hyperflows, a dynamic pruning approach that estimates each weight's importance by observing the network's gradient response to the weight's removal. A global *pressure* term continuously drives all weights toward pruning, with those critical for accuracy being automatically regrown based on their *flow*, the aggregated gradient signal when they are absent. We explore the relationship between final sparsity and pressure, deriving power-law equations similar to those found in neural scaling laws. Empirically, we demonstrate state-of-the-art results with ResNet-50 and VGG-19 on CIFAR-10 and CIFAR-100.

## 1. Introduction

Overparameterization has become the norm in modern deep learning to achieve state-of-the-art performance (Neyshabur et al., 2019; Allen-Zhu et al., 2019; Li et al., 2018). Despite clear benefits for training, this practice also increases computational and memory costs, complicating deployment on resource-constrained devices such as edge hardware, IoT platforms, and autonomous robots (Shi et al., 2016; Li et al., 2019). Recent theoretical and empirical findings suggest that sparse subnetworks extracted from large dense models can match or exceed the accuracy of their dense counterparts (Frankle & Carbin, 2019; Zhou et al., 2019; Ma et al., 2021; Lee et al., 2019; De Jorge et al., 2021; Cho et al., 2023; Yite et al., 2023; Frantar et al., 2024; Wang et al., 2023) and even

outperform smaller dense models of equal size (Ramanujan et al., 2020; Li et al., 2020; Zhu & Gupta, 2018). These results have created interest in network pruning as a strategy to identify minimal, high-performing subnetworks.

Pruning has a rich history (LeCun et al., 1989; Mozer & Smolensky, 1988; Thimm & Hoppe, 1995) and continues to prove valuable for real-time applications (Han et al., 2016; Jongsoo et al., 2017; Wang et al., 2019). Recent methods have significantly advanced the field by resorting to a variety of strategies, from heuristics, gradient methods and Hessian-based criteria (Han et al., 2015; 2016; LeCun et al., 1992; Singh & Alistarh, 2020; Bellec et al., 2018) to dynamic pruning approaches (Liu et al., 2020; Cho et al., 2023; Savarese et al., 2020; Kusupati et al., 2020; Wortsman et al., 2019) or combinations thereof. However, the strong interdependencies between weights remain a challenge (Jin et al., 2020; Templeton et al., 2024; Lee et al., 2019; De Jorge et al., 2021; Louizos et al., 2017), as they complicate the task of determining each weight's absolute importance.

Given this gap, we ask: *Can we rigorously quantify a weight's importance for model accuracy, while accounting for the inherent interrelatedness among neural network weights?*

Inspired by the well-known insight that the value of something is not truly known until it is lost, we introduce *Hyperflows*, a dynamic pruning method which determines weight importance by first removing it. Each weight $\theta_i$ will be pruned if its associated flow parameter $t_i$ is negative. The value of $t_i$ will follow the direction of $|\theta_i|$, as their gradients are strongly correlated, while a global *pressure* term $L_{-\infty}$ will push all $t$ values towards $-\infty$. When an important weight $\theta_i$ is pruned, the network will attempt to increase $t_i$. If the aggregated gradient over multiple iterations of $t_i$, which we call *flow*, is larger than the aggregated pressure, the removed weight will be restored, otherwise it will remain pruned. By allowing this process to happen concurrently on all weights multiple times, the network's topology becomes noisy, disentangling the restoration process from a specific configuration and therefore providing a good approximation for weight importance.

We analyze the relationships between sparsity and pres-

[1]Anonymous Institution, Anonymous City, Anonymous Region, Anonymous Country. Correspondence to: Anonymous Author <anon.email@domain.com>.

Preliminary work. Under review by the International Conference on Machine Learning (ICML). Do not distribute.

sure, obtaining power-law dependencies similar to those of known scaling laws in neural networks (Hestness et al., 2017; Kaplan et al., 2020; Henighan et al., 2020; Rosenfeld et al., 2020; Gordon et al., 2021; Hernandez et al., 2021; Zhai et al., 2021; Hoffmann et al., 2022).

Our method is able to improve over baseline performance for Resnet-50 and VGG-19 on CIFAR-10 for sparsities up to 98%, exceeding current state of the art method and proving the ability of Hyperflows to preserve accuracy.

Summarizing, our key contributions are:

- We introduce *Hyperflows*, a dynamic pruning method aiming to quantify weight importance by developing the notions of *flow* and *pressure*.

- We set a new state-of-the-art benchmark, achieving better accuracy than existing methods in empirical validation, across several networks and datasets.

- We explore the mathematical relationships between pressure and sparsity, finding power-laws similar to those in neural scaling laws.

## 2. Related work

Research on neural network pruning has a relatively old history, with some methods going back decades and laying the groundwork for modern approaches. Early techniques, such as (LeCun et al., 1989) and (LeCun et al., 1992), utilized Hessian-based techniques and Taylor expansions to identify and remove unimportant specific weights, while Mozer & Smolensky (1988) employed derivatives to remove whole units, an early form of structured pruning. These initial studies demonstrated the feasibility of reducing network complexity without significantly compromising performance. An influential overview (Thimm & Hoppe, 1995) concluded that magnitude pruning was particularly effective, a paradigm that since then has been widely adopted (Han et al., 2016; Frankle & Carbin, 2019; Zhou et al., 2019; Evci et al., 2020; Kusupati et al., 2020; Han et al., 2015).

**The existence of highly effective subnetworks** builds upon these foundational theoretical studies, with the Lottery Ticket Hypothesis (Frankle & Carbin, 2019) being a good example. Magnitude pruning is used to demonstrate that there exists a mask which, if applied at the start of training, produces a sparse subnetwork capable of matching the performance of the original dense network after training, if the initialization is kept unmodified. Subsequent research has further validated this concept by showing that these subnetworks produced by masks, even without any training, achieve significantly higher accuracy than random chance (Zhou et al., 2019), reaching up to 80% accuracy on MNIST. Moreover, training these masks instead of the actual weight

values can result in performance comparable to the original network (Ramanujan et al., 2020; Zhou et al., 2019), suggesting that neural network training can occur through mechanisms different from weight updates, including the masking of randomly initialized weights. Other studies have attempted to identify the most trainable subnetworks at initialization. Lee et al. (2019) use gradient magnitudes as a way to identify trainable weights, while Savarese et al. (2020) employ $L_0$ regularization along with a sigmoid function that gradually transitions into a step function during training, enabling continuous sparsification. These findings indicate that the specific values and even the existence of certain weights may be less critical than previously believed.

**Dynamic pruning** differs from classical heuristics by finding sparse networks during training and allowing the model to adjust itself. Some methods use learnable parameters, e.g. Kusupati et al. (2020) train magnitude thresholds for each layer in the network to determine which weights will be pruned. Other works, like that of Cho et al. (2023), do not have any learnable parameters, learning instead a weight distribution whose shape will determine which and how many weights are pruned. Yet another class of $L_0$ regularization techniques (Savarese et al., 2020; Louizos et al., 2018) try to maximize the number of removed weights. Hyperflows aligns with the dynamic pruning paradigm by enabling continuous pruning of weights based on dynamically updated parameters. However, unlike most methods that rely on instantaneous gradients or fixed thresholds, Hyperflows introduces a novel mechanism that assesses each weight's importance through an aggregate gradient signal over multiple iterations.

**Pruning based on gradient values** is another prominent approach, often overlapping with dynamic methods, which enables the assessment of weight properties in relation to the loss function. Lee et al. (2019) and De Jorge et al. (2021) assess the trainability of subnetworks by analyzing initial gradient magnitudes relative to the loss function. AutoPrune (Xiao et al., 2019) introduces handcrafted gradients that influence training, while Dynamic Pruning with Feedback (Lin et al., 2020) uses gradients during backpropagation to recover pruned weights with high trainability, preserving accuracy. Evci et al. (2020) use gradient and weight magnitudes to determine which weights to prune and to regrow. Liu et al. (2022) build upon these concepts, by employing a zero-cost neuroregeneration scheme, which prunes and regrows the same number of weights, effectively keeping the sparsity constant while growing accuracy. Our method uses gradients magnitudes to approximate the importance of a weight when it is pruned, which proves to be effective, since gradients are correlated with the loss of features induced by the pruning that weight. Hyperflows distinguishes itself from other methods by utilizing gradient magnitudes to evaluate the importance of weights after the moment of

their pruning. Instead of predefining which weights are (un)important based solely on instantaneous gradients or single-stage evaluations, Hyperflows identifies a weight's significance based on the aggregated impact its removal has on the network's performance.

## 3. Hyperflows

The main idea of our method is to assess the contribution of individual weights in a neural network by first pruning them and evaluating the resulting impact on performance across various network topologies. This is achieved by introducing a learnable parameter $t_i$ near each weight $\theta_i$ which decides if $\theta_i$ is active or pruned. When a necessary weight is pruned, the gradient of its associated $t_i$, $\frac{\partial \mathcal{L}}{\partial t_i}$, termed weight *flow*, increases $t_i$, which regrows the weight. Flow is strongly correlated with the decrease in performance when the weight is removed, thereby serving as a good approximation for its importance. An $L_{-\infty}$ penalty, defined below in (7), is used to push $t$ values towards pruning and control the overall sparsity. As compression occurs, the remaining weights will have increased importance, by capturing the features lost from the permanently pruned weights, leading to larger flows. We analyze this effect in Appendix B.

### 3.1. Preliminaries

Consider a neural network defined as a function:

$$f : \mathcal{X} \times \theta \to \mathcal{Y},$$

where $\mathcal{X}$ is the input space, $\mathcal{Y}$ is the output space, and $\theta \subseteq \mathbb{R}^d$ denotes the weight vector.

Given a training set $\{(x_j, y_j)\}_{j=1}^J$, learning the parameters $\theta$ amounts to minimizing a loss function:

$$\min_\theta \sum_{j=1}^J \ell\big(f(x_j, \theta), y_j\big),$$

so that $f(x_j, \theta)$ aligns with $y_j$.

We define the topology of the neural network $\mathcal{T}$ as a binary vector $\mathcal{T} \in \{0,1\}^d$ where $\mathcal{T}^i \in \{0,1\}$ represents whether weight $\theta_i$ is pruned or not. We denote a family of topologies as $\{\mathcal{T}_k\}_{k=1}^K$, with $K$ its cardinality. Thus, the loss of a network with topology $\mathcal{T}$ is:

$$\mathcal{L}(\mathcal{T}) = \sum_{j=1}^J \ell\big(f(x_j, \theta \odot \mathcal{T}), y_j\big),$$

where $\odot$ is the Hadamard product. Note that $\mathcal{L}(\mathcal{T})$ depends on $\theta$.

For each parameter $\theta_i$, we introduce a learnable scaler $t_i$ to which we refer as *flow parameter*. We denote with $t$ the

vector of flow parameters. Vector $t$ is used to generate the topology $\mathcal{T}$ with $\mathcal{T}^i = H(t_i)$, where:

$$H(t_i) = \begin{cases} 1 & \text{if } t_i > 0, \\ 0 & \text{if } t_i \le 0. \end{cases}$$

Thus, if $t_i > 0$ then $\theta_i$ is active, otherwise ($t_i \le 0$), $\theta_i$ is pruned. We denote with $\mathcal{T} \setminus \{\theta_i\}$ a topology $\mathcal{T}$ for which we set $\mathcal{T}^i = 0$ and define the change in $\mathcal{L}(\mathcal{T})$ when $\theta_i$ is removed as:

$$\Delta \mathcal{L}(\mathcal{T} \setminus \{\theta_i\}) = \mathcal{L}(\mathcal{T} \setminus \{\theta_i\}) - \mathcal{L}(\mathcal{T}).$$

We use a global penalty term $L_{-\infty}$ to push all $t_i$ values towards $-\infty$, which we discuss in detail in Section 3.2. Our goal is to find a topology $\mathcal{T}_f$ and set of weights $\theta$ such that the following loss is minimal:

$$\mathcal{J}(\mathcal{T}) = \mathcal{L}(\mathcal{T}) + L_{-\infty}(t). \tag{1}$$

### 3.2. Weight Flow

Since the optimal topology $\mathcal{T}^*$ is initially unknown, any metric for the importance of $\theta_i$ measured on the initial topology $\mathcal{T}_0$ might not be relevant for $\mathcal{T}^*$. For this reason, weight importance is evaluated multiple times during training. We present an importance metric, called *flow*, tied to a specific topology $\mathcal{T}$, and then extend it to a family of topologies $\{\mathcal{T}_k\}_{k=1}^K$. We begin by defining the gradient:

$$\mathcal{G}(\theta_i, \mathcal{T}) = \frac{\partial \mathcal{L}(\mathcal{T})}{\partial t_i}, \forall t_i \in \mathbb{R}. \tag{2}$$

Importantly, the sign of the gradient $\mathcal{G}(\theta_i, \mathcal{T})$ always follows the direction of $|\theta_i|$ (proof in Appendix A.2). If $\theta_i$ increases or decreases in magnitude, then $t_i$ will correspondingly increase or decrease.

In our method, $\mathcal{G}(\theta_i, \mathcal{T})$ takes two different meanings based on whether $t_i > 0$ or $t_i \le 0$. We first define the meaning in the case $t_i \le 0$ as *flow* for one topology:

$$\mathcal{F}(\theta_i, \mathcal{T}) = \begin{cases} \mathcal{G}(\theta_i, \mathcal{T}) & t_i \le 0, \\ 0 & t_i > 0. \end{cases} \tag{3}$$

When $\theta_i$ is pruned ($t_i \le 0$) from topology $\mathcal{T}$, a corresponding $\Delta \mathcal{L}(\mathcal{T} \setminus \{\theta_i\})$ will occur. If $\Delta \mathcal{L}(\mathcal{T} \setminus \{\theta_i\}) > 0$, then the pruning of $\theta_i$ leads to an increase in the loss, and increasing $|\theta_i|$ from 0 back to the original will reduce the loss again. Otherwise, if $\Delta \mathcal{L}(\mathcal{T} \setminus \{\theta_i\}) \le 0$, increasing $|\theta_i|$ will not reduce the loss. Therefore, any parameter whose gradient depends on changes in $|\theta_i|$ will have its value increased only if the pruned weight was important. Overall, $\mathcal{F}(\theta_i, \mathcal{T})$ will create large positive changes in $t_i$ for important pruned weights, regrowing them (proof in Appendix A.1).

For the case $t_i > 0$, we define:

$$\mathcal{M}(\theta_i, \mathcal{T}) = \begin{cases} 0 & t_i \leq 0, \\ \mathcal{G}(\theta_i, \mathcal{T}) & t_i > 0. \end{cases} \quad (4)$$

Minimizing the loss function does not inherently correlate with weight magnitude increases. Therefore $\mathcal{M}(\theta_i, \mathcal{T}_k)$ will represent changes in magnitude over training. Pruning will be encouraged for weights whose magnitude decreases, and be resisted for those whose magnitudes are increasing (proof in Appendix A.2).

All the functions involved in backpropagation must be differentiable, but $H$ is not. Since $\mathcal{F}(\theta_i, \mathcal{T})$ should only depend on the importance of $\theta_i$ and not the value of $t_i$, we choose a straight-through estimator for the gradient $\frac{\partial H}{\partial t_i} = 1$, which is also used for $\mathcal{M}(\theta_i, \mathcal{T})$ and therefore for the overall gradient $\mathcal{G}(\theta_i, \mathcal{T})$.

In practice, we analyze weight behaviour over several topologies. Extending equations (3) and (4) to a family of topologies we obtain the aggregated flow and the average change of the weight magnitude respectively:

$$\mathcal{F}(\theta_i, \{\mathcal{T}_k\}_{k=1}^{K}) = \frac{1}{K} \cdot \sum_{k=1}^{K} \mathcal{F}(\theta_i, \mathcal{T}_k), \quad (5)$$

$$\mathcal{M}(\theta_i, \{\mathcal{T}_k\}_{k=1}^{K}) = \frac{1}{K} \cdot \sum_{k=1}^{K} \mathcal{M}(\theta_i, \mathcal{T}_k). \quad (6)$$

To drive $t$ values towards $-\infty$, we employ an "$L_{-\infty}$" loss called *pressure*, formulated as:

$$L_{-\infty}(t) = \frac{1}{d} \cdot \gamma \cdot \sum_{i=1}^{d} t_i, \quad (7)$$

where $\gamma$ is a scalar used to control sparsity and $d$ the number of weights in the network. From this point forward, any reference about an increase or decrease in pressure will refer to an increase or decrease in $\gamma$.

It is important to explore how $\mathcal{L}(\mathcal{T})$ and $L_{-\infty}(t)$ interact with $t$ values in (1). For multiple iterations $R$ of training, we get:

$$\sum_{r=1}^{R} \frac{\partial(\mathcal{L}(\mathcal{T}_r) + L_{-\infty}(t))}{\partial t_i} = \sum_{r=1}^{R} (\mathcal{G}(\theta_i, \mathcal{T}_r) + \frac{\gamma}{d}), \quad (8)$$

where $\mathcal{T}_r$ is the topology at iteration $r$. At each iteration, a weight can either be pruned or active, therefore, we can partition a weight's state during training between pruned and active stages. A stage $S_f, f \in \{1, \ldots, F\}$ is a series of consecutive iterations for which our weight is in the same state. Each $S_f$ has a duration of $D_f$, starting at iteration $s_f$ and ending at $e_f$. We denote by $S_f^+$ the stages when a weight is present and $S_f^-$ those when the weight is pruned.

We define gradients taking place during a stage $S_f^+$ and $S_f^-$ as:

$$\nabla S_f^+ = \sum_{r=s_f}^{e_f} \left( \mathcal{M}(\theta_i, \mathcal{T}_r) + \frac{\gamma}{d} \right), \quad (9)$$

$$\nabla S_f^- = \sum_{r=s_f}^{e_f} \left( \mathcal{F}(\theta_i, \mathcal{T}_r) + \frac{\gamma}{d} \right). \quad (10)$$

Note that it is not possible to have two consecutive stages with the same weight state and all weights are present at the start of training. We partition the set of all stages into two $\{S_1^+, S_3^+, \ldots\}$ and $\{S_2^-, S_4^-, \ldots\}$. We refer to the transition between stages as *implicit regrowth*. We do not know in which partition the final stage $S_F$ will be until training ends. Equation (8) becomes:

$$\sum_{r=1}^{R} (\mathcal{G}(\theta_i, \mathcal{T}_r) + \frac{\gamma}{d}) = \nabla S_1^+ + \nabla S_2^- + \ldots + \nabla S_F^{\{+\text{or}-\}}. \quad (11)$$

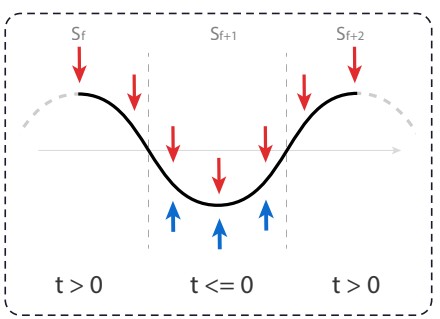

Figure 1. A weight's state can be partitioned into pruned ($t_i > 0$) and active ($t_i \leq 0$) stages. We represent with blue arrows the flow, which appears only for pruned stages, and with red lines the pressure, which appears for all stages.

The partition in stages is illustrated in Figure 1. Analyzing what happens at an individual level for each $\nabla S_i$, by rearranging (9) and (10), we get:

$$\nabla S_f^+ = D_f \cdot \left( \mathcal{M}(\theta_i, \{\mathcal{T}_r\}_{r=s_f}^{e_f}) + \frac{\gamma}{d} \right), \quad (12)$$

$$\nabla S_f^- = D_f \cdot \left( \mathcal{F}(\theta_i, \{\mathcal{T}_r\}_{r=s_f}^{e_f}) + \frac{\gamma}{d} \right). \quad (13)$$

For $S_f^-$, if $\mathcal{F}(\theta_i, \{\mathcal{T}_r\}_{r=s_f}^{e_f}) + \frac{\gamma}{d} > 0$, then the overall change in $t_i$ will be negative, keeping the weight pruned. Otherwise, if $\mathcal{F}(\theta_i, \{\mathcal{T}_r\}_{r=s_f}^{e_f}) + \frac{\gamma}{d} < 0$ then $t_i$ will be increased. In other words, all weights will be pushed towards pruning by the pressure and regrown if the flow is greater than the pressure.

Weights can be deemed unimportant in $\mathcal{T}_r$ and become relevant later in $\mathcal{T}_{r+q}$. However, if a weight is pruned at $\mathcal{T}_r$, the

pressure will continue to push its $t$ value towards $-\infty$ for $q+1$ iterations, making it difficult to regrow. To control this effect, in practice we apply the loss only on $t$ values which are above a certain threshold $T$.

$$\widehat{L}_{-\infty}(t) = \gamma \cdot \sum_{i=1}^{d} t_i \cdot H(T - t_i). \quad (14)$$

### 3.3. Neural pruning laws

We investigate how the pruning pressure scaler $\gamma$, the number of training epochs, and the network architecture shape the evolution of sparsity. These insights lay the foundation for our $\gamma$ scheduler, introduced later, that can reach a target sparsity in any desired training time.

**(0) Sparsity Convergence for a Fixed** $\gamma$. As sparsity increases, the overall flow of the weights will become larger. We ask the following question: Given a fixed $\gamma$, will the network converge to a final sparsity $s$? Moreover, does this mapping from $\gamma$ to $s$ follow any relationship? In Figure 3, we test the existence of convergence empirically by running LeNet-300 on MNIST and ResNet-50 on Cifar-10. We allow each network to train for 300 to 1000 epochs with a constant $\gamma$ pressure and observe the results. We do this with two different optimizers for $t$ values, SGD and Adam. Our findings suggest that there is no one curve that fits the decrease in parameters for both optimizers, but the final convergence point is the same regardless of the optimizer used.

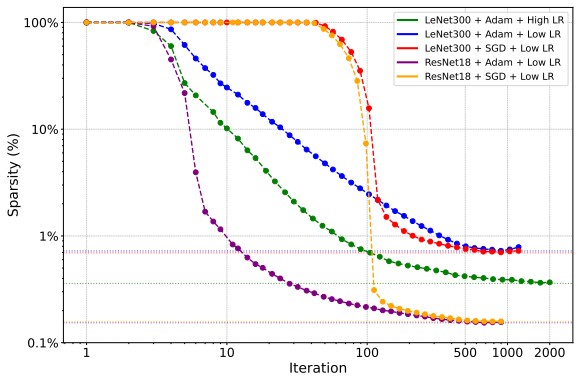

*Figure 2.* Convergence for fixed $\gamma$. We can observe that for each case there is a certain point at which weights are not pruned anymore or offer an extremely high amount of resistance

An important observation is that the final convergence point $s_c$ is influenced by the $\theta$ learning rate $\eta$. If $\eta$ is high, convergence happens in a larger number of epochs (1000 in our experiments), at a higher sparsity. If $\eta$ is low, convergence happens sooner 300 epochs, to a lower sparsity.

**(1) Relationship Between** $\gamma$ **and Final Sparsity**. Assuming that all networks have a sparsity they converge to for a fixed $\gamma$, is there a relationship between $\gamma$ and its associated final sparsity? Can we predict for a new $\gamma$ the final sparsity a network will converge to? We modify the previous experiment, to run the networks 300 epochs for several values of $\gamma$ between $2^{-15}$ and $2^{10}$. Our empirical results suggest a power-law relationship:

$$\ln(s) = \ln(c) - \alpha_0 \cdot \ln(\gamma) - \alpha_1 \cdot (\ln(\gamma))^2, \quad (15)$$

where constants, c, $\alpha_0$, $\alpha_1$ depend on dataset and network architecture.

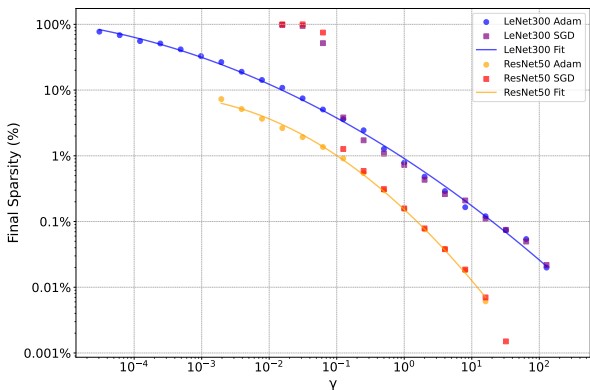

*Figure 3.* Relation between $\gamma$ and final sparsity, showing several curves that can be fitted by our power-law formula. Notice how different optimzers converge to the same points. One particularity of SGD is that for low values of pressure the networks takes longer to converge, which is why a few outliers appear in the top part diagram.

### 3.4. Pressure Scheduler

Our findings from Section 3.3 suggest that for any sparsity we desire, there will be a certain fixed $\gamma$ which produces that sparsity after a fixed number of epochs. However, in practical applications, this $\gamma$ cannot be known from the start, since it would require running the method several times to find out the power-law curve. To solve this issue, we propose a dynamic scheduler that adjusts $\gamma$ at each epoch, driving the network along a desired sparsity trajectory.

The goal of our scheduler, given a function that maps each iteration to a sparsity $f(e) : [0, R] \to [0, 100]$, is to adjust $\gamma$ such that the sparsity of the network after $e$ epochs $s(e)$ will be equal to the desired sparsity $s(e) = f(e)$. Since epochs increase linearly while the relationship between $\gamma$ and sparsity is non-linear, we need our adjustments to $\gamma$ at each epoch to also be non-linear.

We choose $\gamma = p^{\alpha}$ as our $\gamma$ function, where $p$ is adjusted as described in Algorithm 1, $\alpha$ is chosen as a hyperparameter

and $u$ is a constant. We find values of $\alpha \in [1.5, 2.0]$ to be well suited for both stable and accurate pruning. We use inertia terms $p_+$ and $p_-$ to account for the need of potentially larger changes in $\gamma$ for small $\alpha$ values. Our scheduler is able to reach the desired sparsity within a 10% margin of error. Ideally, the training time should be infinite and the changes in $\gamma$ as small as possible, to allow for more controlled pruning. In practice, we find optimal training time for pruning to be somewhere between $T/2$ and $2T$, where $T$ is the original training time needed for the network to converge.

---

**Algorithm 1** Pressure Scheduler

---

1: **Input:** Epoch $e$, sparsity curve $f(e)$, current sparsity $s(e)$, update $u$ (constant), $\alpha$
2: **Internals:** positive inertia $p_+$, negative inertia $p_-$, both initialized with 0.
3: **if** $f(e) < s$ **then**
4:     $p \leftarrow p + u + p_+$
5:     $p_+ \leftarrow p_+ + \frac{u}{4}$
6:     $p_- \leftarrow 0$
7: **else if** $f(e) > s$ **then**
8:     $p \leftarrow p - u - p_-$
9:     $p_- \leftarrow p_- + \frac{u}{4}$
10:     $p_+ \leftarrow 0$
11: **end if**
12: **Return**: pressure $\gamma = p^{\alpha}$

---

### 3.5. Regrowth Stage

One of the main features of our method is the noise created by removal and regrowth of weights, which leads to the disentangling of weight from specific topology. However, this noise is harmful for convergence. For this reason, we introduce a regrowth-only stage at the end, whose purpose is to allow the weights to converge as well as to stabilize the network topology. Specifically, we eliminate the regularization term, setting $\gamma$ to 0 and therefore allowing only weight regrowth. In order to limit the number of regrown weights and add only the most important lost weights, we introduce a decay of $d^e$ for the learning rate $\eta_t$, where $d$ represents a constant and $e$ represents the epoch number.

Despite the number of parameters regrown being hard to control, we can adjust the flow parameters learning rate $\eta_t$, as well as the decay $d$ to constraint or promote the reactivation of weights. Although the absolute number of parameters being regrown is small, the gains in accuracy are significant. For example, we can gain up to 8% accuracy on Imagenet dataset with a net increase in remaining parameters of 0.5%, from 4.0% to 4.5%. Generally, the regrowth phase should be scheduled for between one-quarter to one-third of the total training time to allow adequate reactivation

and stabilization of essential weights.

## 4. Experimental Results

We conducted experiments with *Hyperflows* to demonstrate its effectiveness in achieving high sparsity levels while maintaining accuracy across various neural network architectures and datasets. Since *Hyperflows* quantifies the importance of each weight, we evaluate it for post-training pruning scenario, as weights need to hold significance within the network before starting the pruning process.

We compare *Hyperflow* with state-of-the-art pruning methods such as GraNet (Liu et al., 2022), RigL (Evci et al., 2020), GMP (Trevor Gale, 2019) and Synflow (Hidenori et al., 2020). We run GraNet and GMP individually using two setups. In the first setup, we use their reported best configurations for training networks from scratch. In the second setup, we initialize them with the same baseline network as ours and determine the optimal learning rate for this specific setting. We generally observe that their reported learning rates are also optimal for the post-training scenario. We use the same training budget of 160 epochs and keep all other configurations intact, to ensure no unintentional degradation occurs. We mark all the methods run individually with $*$. For the methods we do not run, we use the results reported in (Liu et al., 2022) and (Kusupati et al., 2020).

We evaluate Hyperflows on the following combinations of networks and datasets: LeNet-300 on MNIST, ResNet-50 on CIFAR-10/100, VGG19 on CIFAR-10/100 and ResNet-50 on ImageNet-1K. Details on the training setups, architectures and datasets are summarized in Appendix D. Unless otherwise stated, all experiments were conducted three times, with results expressed as mean $\pm$ standard deviation. The experiments were conducted on a system equipped with 3 RTX 4090 GPUs. Ablation studies are presented in Appendix B. Our findings indicate that Hyperflows consistently outperforms state-of-the-art pruning methods, achieving higher sparsity levels with comparable or better accuracy across multiple datasets and architectures.

### 4.1. CIFAR-10 / 100

We evaluate the performance of *Hyperflows* on CIFAR-10 and CIFAR-100 using ResNet-50 and VGG-19 architectures. CIFAR-10 is a simpler benchmark with fewer classes, making it a smaller challenge compared to CIFAR-100, which has a larger number of classes and fewer images per class. This makes CIFAR-100 more prone to instability during training and a more rigorous test for pruning methods. Results are presented in Table 1.

A key feature of *Hyperflows* is the intentional introduction of noise during pruning. While this leads to performance fluctuations, it enhances resilience to pruning and supports

*Table 1.* Comparison of classification accuracy (%) on CIFAR-10 and CIFAR-100 datasets at different pruning ratios (90.0%, 95.0%, 98.0%). Results are reported for VGG-19 and ResNet-50 architectures using various pruning methods, including Hyperflows. Bold values represent the best performance for each setting. We consider significant results to have at least 0.25% accuracy difference from the other methods. Otherwise, we will report multiple best performing methods if it is the case.

| Dataset | CIFAR-10 | | | CIFAR-100 | | |
|---|---|---|---|---|---|---|
| Pruning ratio | 90.0% | 95.0% | 98.0% | 90.0% | 95.0% | 98.0% |
| **VGG-19** (Dense) | | **93.85 ± 0.06** | | | **73.44 ± 0.09** | |
| SNIP | 93.63 | 93.43 | 92.05 | 72.84 | 71.83 | 58.46 |
| GraSP | 93.30 | 93.04 | 92.19 | 71.95 | 71.23 | 68.90 |
| STR | 93.73 | 93.27 | 92.21 | 71.93 | 71.14 | 69.89 |
| SIS | 93.99 | 93.31 | 93.16 | 72.06 | 71.85 | 71.17 |
| SynFlow | 93.35 | 93.45 | 92.24 | 71.77 | 71.72 | 70.94 |
| RigL | 93.38±0.11 | 93.06±0.09 | 91.98±0.09 | 73.13±0.28 | 72.14±0.15 | 69.82±0.09 |
| GMP* | 93.82 ± 0.15 | 93.84 ± 0.14 | 92.34 ± 0.13 | 73.57 ± 0.20 | 73.39 ± 0.11 | 72.78 ± 0.07 |
| GraNet* ($s_i = 0$) | 93.87 ± 0.05 | 93.84 ± 0.16 | **93.87 ± 0.11** | 74.08 ± 0.10 | 73.86 ± 0.04 | **73.00 ± 0.18** |
| Hyperflows (ours) | **94.05 ± 0.17** | **94.15 ± 0.14** | **93.95 ± 0.18** | **74.37 ± 0.21** | **74.18 ± 0.15** | 72.9 ± 0.05 |
| **ResNet-50** (Dense) | | **94.72 ± 0.05** | | | **78.32 ± 0.08** | |
| SNIP | 92.65 | 90.86 | 87.21 | 73.14 | 69.25 | 58.43 |
| GraSP | 92.47 | 91.32 | 88.77 | 73.28 | 70.29 | 62.12 |
| STR | 92.59 | 91.35 | 88.75 | 73.45 | 70.45 | 62.34 |
| SIS | 92.81 | 91.69 | 90.11 | 73.81 | 70.62 | 62.75 |
| SynFlow | 92.49 | 91.22 | 88.82 | 73.37 | 70.37 | 62.17 |
| RigL | 94.45±0.43 | 93.86±0.25 | 93.26±0.22 | 76.50±0.33 | 76.03±0.34 | 75.06±0.27 |
| GMP* | 94.81 ± 0.05 | 94.89 ± 0.1 | 94.52 ± 0.12 | 78.39 ± 0.18 | 78.38 ± 0.43 | 77.16 ± 0.25 |
| GraNet* ($s_i = 0$) | 94.69 ± 0.08 | 94.44 ± 0.01 | 94.34 ± 0.17 | 79.09 ± 0.23 | 78.71 ± 0.16 | **78.01 ± 0.20** |
| Hyperflows (ours) | **95.41 ± 0.12** | **95.15 ± 0.11** | **95.26 ± 0.13** | **79.58 ± 0.18** | **79.23 ± 0.16** | 77.7 ± 0.08 |

an effective regrowth phase, achieving state-of-the-art accuracy. Further analysis of these fluctuations is provided in Appendix B.

On CIFAR-10, *Hyperflows* achieves above-baseline performance for all sparsity levels (10%, 5%, and 2%) for both VGG-19 and ResNet-50, making it the only method to do so. Accuracy differences between *Hyperflows* and the next best method are generally within 1% or less. For VGG-19, *Hyperflows* outperforms GraNet* at 90% sparsity by 0.18% and GMP* by 0.23%. At 98% sparsity, the difference with GMP* increases to 1.61%. For ResNet-50, *Hyperflows* maintains a consistent 0.7% accuracy advantage over GraNet* across all sparsity levels. Additionally, we study layer-wise sparsity for ResNet-50 on CIFAR-10 at extreme sparsity levels (99.74%, 99.01%, 98.13%) and analyze weight distributions under extreme compression. We observe that exreme sparsity significantly alters the distribution of weights, results are detailed in Appendix C.1.

On CIFAR-100, *Hyperflows* achieves the best performance in 4 out of 6 benchmarks. In the remaining 2 cases, it is slightly behind GraNet*, with differences of 0.1% and 0.3%. Notably, GraNet* benefits significantly from loading the

baseline network first, gaining nearly 2% accuracy points for ResNet-50 compared to their reported results. Hyperflows outperforms all methods, including GraNet*, at 90% and 95% sparsity by 0.5% accuracy points. Other methods, such as SIS (Verma & Pesquet, 2021), STR (Kusupati et al., 2020), GraSP (Chaoqi Wang & Grosse, 2020), and SynFlow (Hidenori et al., 2020), suffer significant accuracy degradation at 98% sparsity, dropping to 70% ± 2 for VGG-19 and 62% ± 0.5 for ResNet-50.

## 4.2. ImageNet-2012

To evaluate the scalability and effectiveness of *Hyperflows* in large-scale settings, we conducted experiments on the ImageNet-2012 dataset using the ResNet-50 architecture. The complexity and size of this dataset provide a rigorous test for pruning methods, especially at high sparsity levels. Table 2 summarizes the comparison between *Hyperflows* and other prominent pruning techniques, including GraNet (Liu et al., 2022), RigL (Evci et al., 2020), GMP (Trevor Gale, 2019), and STR (Kusupati et al., 2020). Notably, *Hyperflows* maintains higher accuracy across extreme sparsity levels, demonstrating its robustness and effective-

ness in preserving model performance despite significant pruning.

Table 2. Comparison of Top-1 accuracy (%), number of parameters (Params), and sparsity levels (%) for ResNet-50 on the ImageNet-2012 dataset using various pruning methods, including GMP, DNW, RigL, GraNet, STR, and *Hyperflows*. Results are reported at sparsity levels of 90%, 95%, and 96.5%.

| Method | Top-1 Acc (%) | Params | Sparsity (%) |
|---|---|---|---|
| ResNet-50 | 77.01 | 25.6M | 0.00 |
| GMP | 73.91 | 2.56M | 90.00 |
| DNW | 74.00 | 2.56M | 90.00 |
| RigL | 73.00 | 2.56M | 90.00 |
| GraNet | 74.50 | 2.56M | 90.00 |
| STR | 74.31 | 2.49M | 90.23 |
| Hyperflows | 74.40 | 2.54M | 90.11 |
| GMP | 70.59 | 1.28M | 95.00 |
| DNW | 68.30 | 1.28M | 95.00 |
| GraNet | 72.30 | 1.28M | 95.00 |
| RigL | 70.00 | 1.28M | 95.00 |
| STR | 70.40 | 1.27M | 95.03 |
| Hyperflows | 72.44 | 1.13M | 95.58 |
| RigL | 67.20 | 0.90M | 96.50 |
| STR | 67.22 | 0.88M | 96.53 |
| GraNet | 70.5 | 0.90M | 96.50 |
| Hyperflows | 70.91 | 0.92M | 96.42 |

At 96.5% sparsity, *Hyperflows* achieves a Top-1 accuracy of 70.91%, surpassing RigL and STR by 3% accuracy, while GraNet has similar performance at 70.51%. At 95% sparsity, *Hyperflows* achieves a Top-1 accuracy of 72.14%, significantly outperforming GMP (70.59%), DNW (68.30%), RigL (70.00%), and STR (70.40%), while GraNet falls behind by 0.14%. At 90% sparsity, Hyperflows achieves 74.40% accuracy closely matching GraNet (74.50%) and STR (74.31%).

Furthermore, we conducted an analysis on the weight histograms of ResNet-50 on ImageNet to study the difference in weight distribution under two settings measured at same sparsities during pruning and regrowth stages. The results are shown in Appendix C.1, where we observe a shift in the weight distributions and a decrease in the number of non-zero weights during the pruning phase.

### 4.3. LeNet-300

We utilize LeNet-300 for our experiments due to its simplicity and manageability, which make it an ideal choice for examining various aspects of our method. By applying our *flow* metric to eliminate non-essential weights, we progressively increase the sparsity of LeNet-300 up to 99.85% and

investigate which remaining weights in the first layer are crucial for classification. As shown in Figure 4, the areas essential for classification become evident, with the margin being pruned across all levels of sparsity. The dataset uses only normalization, with no additional transformations applied. Additionally, in Appendix C.2, we track the number of weight flips per iteration during training, and visualize how gradient flow contributes to reactivating important weights.

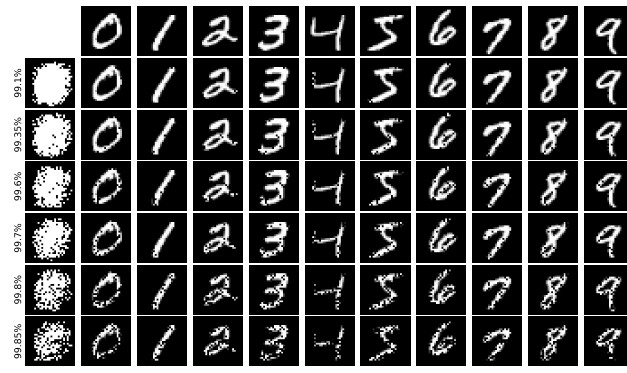

Figure 4. The pixels utilized during inference for various sparsified LeNet-300 networks. The first column shows the masks, with white pixels indicating those used at inference, while the first row presents the original, unmasked images.

## 5. Discussion & Conclusion

We introduced Hyperflows, a theoretical framework around the idea of weight importance along with the notions of pressure and flow. We studied the relationships between flow, pressure and the final sparsity of a neural network, which we termed neural pruning laws. Based on these laws, we developed a pressure scheduler, allowing us to indirectly control sparsity, as opposed to pruning or regrowing a fixed number of weights. Furthermore, we achieved state-of-the-art results on benchmarks such as CIFAR-10 and CIFAR-100, overall demonstrating the potential of Hyperflows, from both an empirical and theoretical perspective. In future work, we aim to explore whether the constants in the pruning laws exhibit any properties that hold across multiple networks and datasets. Furthermore, it would be interesting to adapt Hyperflows to other problems and architectures, such as Reinforcement Learning and Large Language Models.

## Impact Statement

This work's dynamic pruning approach can significantly reduce the computational and energy costs of deep learning models, making large networks more efficient and accessible for a wider range of applications. By achieving extreme sparsity with minimal accuracy loss, it could enable real-

time or low-resource usage in domains like healthcare or edge AI, while also lowering the overall environmental impact of machine learning.

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

## A. Analysis

### A.1. The Role of Weight Importance in Generating Stronger Flows

In this section, we provide theorethical groundings for the correlation between weight importance and the magnitude of the corresponding flow within the Hyperflows framework. To this end, i.e. we analyse why important weights induce stronger flows within the Hyperflows framework, we analyze the relationship between weight importance and the resulting gradient feedback. Specifically, we show that if pruning a weight $\theta_i$ leads to a larger increase in the loss function compared to pruning another weight $\theta_j$, then the flow of $\theta_i$ is larger than that of $\theta_j$.

To assess the significance of a specific weight, we measure the change in loss induced by pruning the weight. Formally, a weight $\theta_i$ is considered more important than a weight $\theta_j$ if:

$$\Delta\mathcal{L}(\mathcal{T} \setminus \{\theta_i\}) > \Delta\mathcal{L}(\mathcal{T} \setminus \{\theta_j\}), \quad \text{where} \quad \Delta\mathcal{L}(\mathcal{T} \setminus \{\theta_i\}) = \mathcal{L}(\mathcal{T} \setminus \{\theta_i\}) - \mathcal{L}(\mathcal{T}).$$

Note that we make the assumption that all weights have a positive importance value, i.e. $\Delta\mathcal{L}(\mathcal{T} \setminus \{\theta_i\}) > 0$.

$$\mathcal{F}(\theta_i, \mathcal{T}) = \begin{cases} \frac{\partial\mathcal{L}(\mathcal{T})}{\partial t_i} & \text{if } t_i \leq 0, \\ 0 & \text{otherwise.} \end{cases} \tag{16}$$

**Proposition A.1.** *If $\Delta\mathcal{L}(\mathcal{T} \setminus \{\theta_i\}) > \Delta\mathcal{L}(\mathcal{T} \setminus \{\theta_j\})$, then the flow $\mathcal{F}(\theta_i, \mathcal{T})$ exceeds $\mathcal{F}(\theta_j, \mathcal{T})$ in magnitude:*

$$\left| \frac{\partial\mathcal{L}(\mathcal{T})}{\partial t_i} \right| > \left| \frac{\partial\mathcal{L}(\mathcal{T})}{\partial t_j} \right|.$$

*Proof.* Consider two weights $\theta_i$ and $\theta_j$ with flow parameters $t_i$ and $t_j$, respectively. Let $\delta > 0$ be a small perturbation applied to the flow parameters such that pruning a weight involves decrementing its parameter by $\delta$, ensuring $t_k - \delta \leq 0$ for $k \in \{i, j\}$. This results in the topology changes $\mathcal{T} \setminus \{\theta_i\}$ and $\mathcal{T} \setminus \{\theta_j\}$.

We analyze how the pruning of each of the two weights affects the loss function $\mathcal{L}(\mathcal{T})$:

1. **Pruning $\theta_i$:** When $\theta_i$ is pruned, the topology changes to $\mathcal{T} \setminus \{\theta_i\}$. Using a first-order Taylor expansion around $\mathcal{T}$, the change in loss due to pruning $\theta_i$ is approximated by:

$$\mathcal{L}(\mathcal{T} \setminus \{\theta_i\}) - \mathcal{L}(\mathcal{T}) \approx -\delta \cdot \frac{\partial\mathcal{L}(\mathcal{T})}{\partial t_i} + \mathcal{O}(\delta^2).$$

2. **Pruning $\theta_j$:** Similarly, pruning $\theta_j$ results in the topology $\mathcal{T} \setminus \{\theta_j\}$ and the corresponding change in loss due to pruning $\theta_j$ is approximated by:

$$\mathcal{L}(\mathcal{T} \setminus \{\theta_j\}) - \mathcal{L}(\mathcal{T}) \approx -\delta \cdot \frac{\partial\mathcal{L}(\mathcal{T})}{\partial t_j} + \mathcal{O}(\delta^2).$$

Given the assumption that pruning $\theta_i$ induces a larger increase in loss compared to pruning $\theta_j$:

$$\Delta\mathcal{L}(\mathcal{T} \setminus \{\theta_i\}) > \Delta\mathcal{L}(\mathcal{T} \setminus \{\theta_j\}),$$

substituting the approximations yields:

$$\left| -\delta \cdot \frac{\partial\mathcal{L}(\mathcal{T})}{\partial t_i} \right| > \left| -\delta \cdot \frac{\partial\mathcal{L}(\mathcal{T})}{\partial t_j} \right|.$$

Since $\delta > 0$ is a constant, this simplifies to:

$$\left| \frac{\partial\mathcal{L}(\mathcal{T})}{\partial t_i} \right| > \left| \frac{\partial\mathcal{L}(\mathcal{T})}{\partial t_j} \right|.$$

Thus, the above inequality directly translates to:

$$|\mathcal{F}(\theta_i, \mathcal{T})| > |\mathcal{F}(\theta_j, \mathcal{T})|.$$

$\square$

## A.2. Influence of Weight Magnitude and Direction on Flow Parameters

To observe the relationship between the magnitude and direction of $\theta_i$ and its flow parameter $t_i$, $t_i > 0$, we analyze the partial derivatives of the loss function $\mathcal{L}$ with respect to both $\theta_i$ and $t_i$. We consider the following derivatives:

$$\frac{\partial \mathcal{L}}{\partial t_i} = \frac{\partial \mathcal{L}}{\partial \alpha} \cdot \frac{\partial \alpha}{\partial H} \cdot \frac{\partial H}{\partial t_i} = \frac{\partial \mathcal{L}}{\partial \alpha} \cdot \theta_i \cdot \mathcal{I} \cdot 1,$$

$$\frac{\partial \mathcal{L}}{\partial \theta_i} = \frac{\partial \mathcal{L}}{\partial \alpha} \cdot \frac{\partial \alpha}{\partial \theta_i} = \frac{\partial \mathcal{L}}{\partial \alpha} \cdot H(t_i) \cdot \mathcal{I}$$

where $\alpha = \theta_i \cdot H(t_i) \cdot \mathcal{I}$ and $H(t_i) = 1$.

**Proposition A.2.** *The sign of $\frac{\partial \mathcal{L}(\mathcal{T})}{\partial t_i}$ is influenced by both the gradient of the loss with respect to $\theta_i$ and the value of $\theta_i$. Specifically, the update direction of $t_i$ is determined by the product $\theta_i \cdot \frac{\partial \mathcal{L}(\mathcal{T})}{\partial \theta_i}$.*

*Proof.* From the expression for $\frac{\partial \mathcal{L}(\mathcal{T})}{\partial t_i}$, we observe that:

$$\frac{\partial \mathcal{L}}{\partial t_i} = \theta_i \cdot \frac{\partial \mathcal{L}}{\partial \theta_i}, \forall t_i > 0.$$

Table 3 showcases the implications based on the sign and magnitude of $\theta_i$. $\square$

*Table 3.* Implications Based on the Sign of $\theta_i$

| Weight Type | Derivative $\frac{\partial \mathcal{L}}{\partial \theta_i}$ | Change in $\theta_i$ | Gradient $\frac{\partial \mathcal{L}}{\partial t_i}$ | Effect on $t_i$ | Implication |
|---|---|---|---|---|---|
| $\theta_i > 0$ | Positive | $\theta_i \downarrow$ | Positive | $t_i \downarrow$ | Reinforces pruning |
| $\theta_i > 0$ | Negative | $\theta_i \uparrow$ | Negative | $t_i \uparrow$ | Promotes regrowth |
| $\theta_i < 0$ | Positive | $|\theta_i| \uparrow$ | Positive | $t_i \uparrow$ | Promotes regrowth |
| $\theta_i < 0$ | Negative | $|\theta_i| \downarrow$ | Negative | $t_i \downarrow$ | Reinforces pruning |

# B. Ablation Studies

## B.1. Factors influencing flow $\mathcal{F}(\theta_i, \mathcal{T})$

We begin by analyzing how the flow value $\mathcal{F}(\theta_i, \mathcal{T})$ is influenced by factors other than the learning rate $\eta_t$. Our findings from Section 3.3, suggest that weight learning affects the behavior of flow, by changing the final convergence point a network will reach for the same constant pressure $\gamma$. We study this effect in the case of LeNet-300. We run the network for 1000 epochs for three different learning rates of $0.005, 0.0005$ and $0.00005$, with no schedulers used and the same constant $\gamma$. Our findings are summarized in Figure 5, which shows that increasing weight learning rate $\eta_\theta$ leads to smaller flows and convergence at higher sparsities.

Given the impact of $\theta$ learning rate on network convergence, we study the influence of high and low learning rates on our pruning and regrowth phases. In our experiments, we study three setups on ResNet-50 CIFAR-10. In the first two experiments, we study how constant learning rates across the entire pruning and regrowth process affect sparsity and regrowth. We choose a high learning rate of $0.01$ and a low learning rate of $0.0001$. For our third experiment, we start with the high learning rate which is then decayed using cosine annealing to a low learning rate until the end of regrowth. For all three studies we let our scheduler guide the network towards the same sparsity rate of $1\%$. However, we observe significant differences in the regrowth stage. For the first experiment, regrowth does not occur at all, with more weights being pruned even after the pressure is set to 0, while

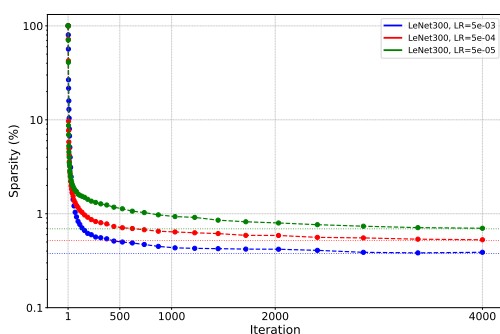

*Figure 5.* MNIST convergence for constant $\gamma = 1$ for different learning rates

for the low learning rate, the performance initially degrades, but is followed by a substantial regrowth stage where the number of remaining parameters increases by $60\%$. For the third experiment performance does not degrade as much as for the low learning rate and the regrowth is done in a more controlled way, experiencing an increase in remaining parameters of $35\%$. The results are illustrated in Figure 6.

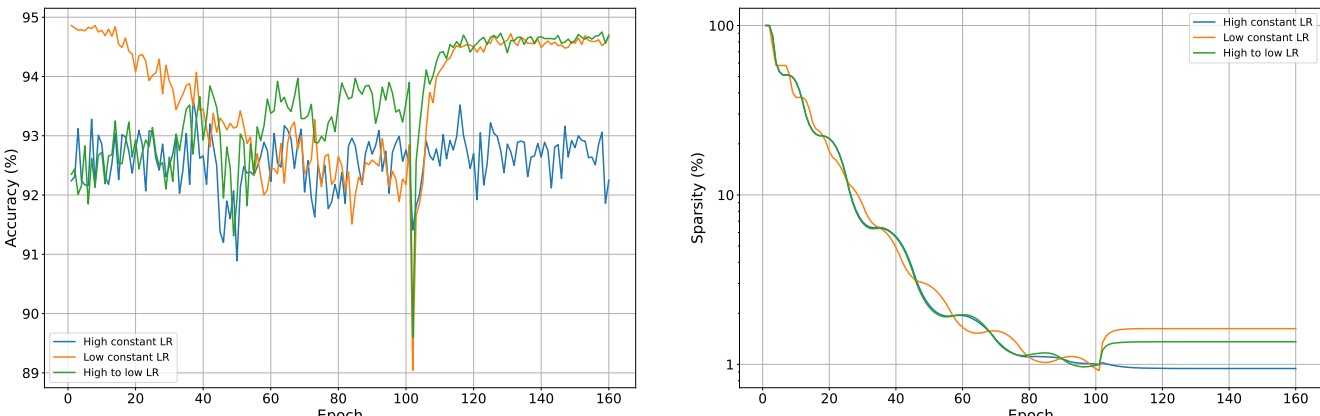

*Figure 6.* The impacts of weights learning rate on pruning and sparsity

Lastly, we study how weight flow is affected by weight decay. Being directly applied on the weights, weight decay acts on both pruned and present weights. If a weight has been pruned in the first epochs on the training, weight decay will keep making it smaller and smaller, in this way diminishing its flow. We run similar experiments to the ones before, with a learning rate of $0.01$, decayed during training to $0.0001$, both with and without the standard weight decay. As expected, we observe in Figure 7 that regrowth without weight decay if more ample. We run this experiment five times, and note that each time the pattern illustrated in the figure remain consistent.

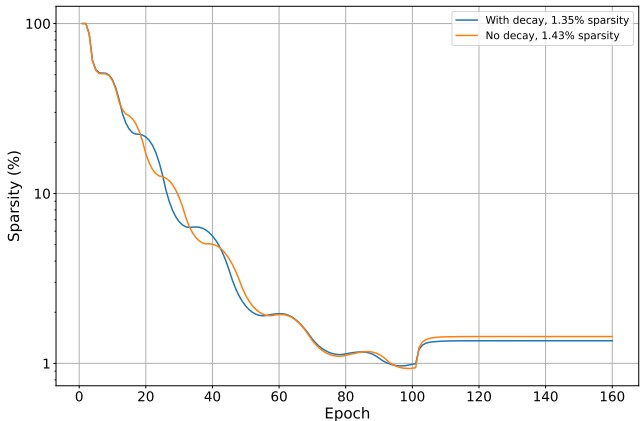

*Figure 7.* Effect of weight decay on the regrowth process

### B.2. Weights $\eta_\theta$ and pruning

Given the large impact $\eta_\theta$ has on flow, we explore its implications for producing an optimal pruning setup for Hyperflows. We run three experimental setups on ResNet-50 CIFAR-10 similar to the ones before. For each one of them, we select a starting learning rate, which is then decayed during training to $0.0001$ to ensure convergence. For this setup, we run experiments using $\eta_\theta = 0.1, 0.01, 0.0001$. We analyze the results from the perspective of accuracy after pruning, noise, regrowth and final accuracy. We find that the third setup is the most effective for Hyperflows.

We observe that each of the four studied aspects has a relationship with the learning rate. The noise is increased as initial learning rate increases, accuracy at the end of pruning is decreased the most for low learning rates and the highest for large

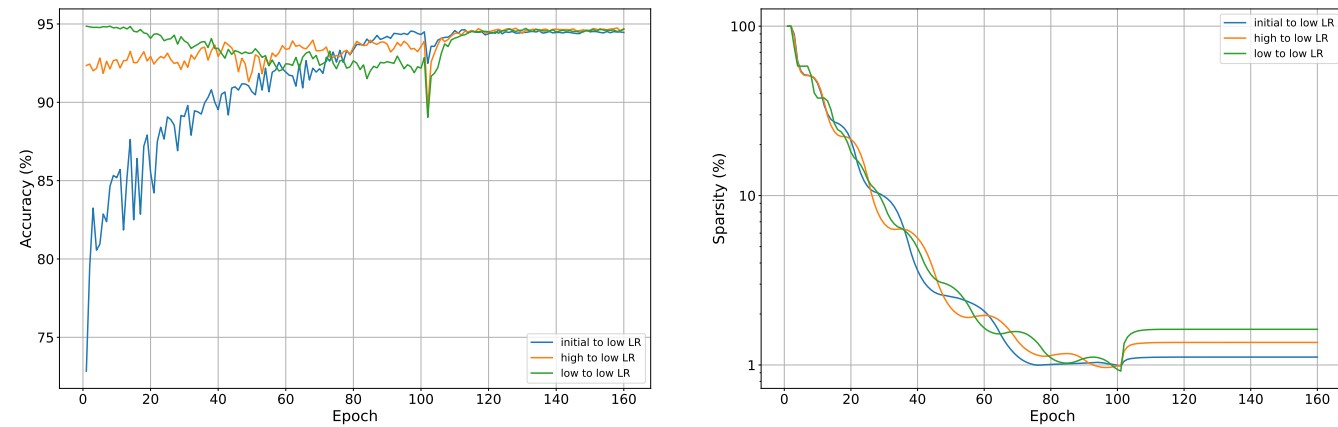

*Figure 8.* Training evolution for different learning rate configurations.

learning rates. We obtain the highest final accuracy for higher learning rates and the regrowth phase is diminished the higher the learning rate. These relationships hold and can be easily seen in Figure 8

### B.3. $\eta_t$ values and regrowth

We analyze regrowth behavior for several values of $\eta_t$. At regrowth stage, we scale $\eta_t$ with $5, 10, 20, 30$ for VGG-19 CIFAR-100 to observe the behavior of regrowth stage. Our findings are summarized in Figure 9. As $\eta_t$ increases so does the number of regrown weights. However, we note that after a point, generally about an increase of $50\%$ in remaining parameters, the effects of regrowth start to be diminished and starts introducing noise in the performance, while also regrowing more weights.

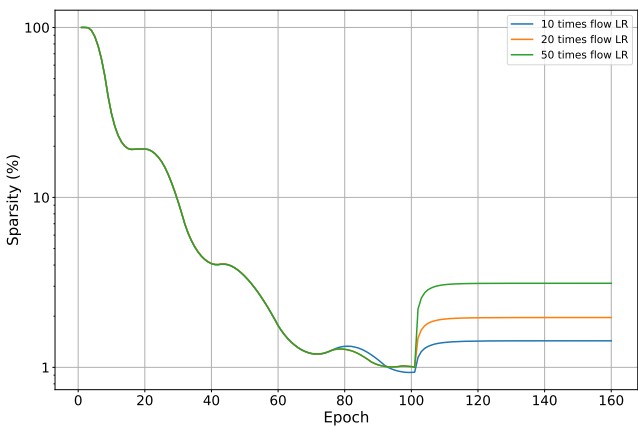

*Figure 9.* How differently scaled $\eta_t$ affect regrowth

## C. Extended experiments

### C.1. Layerwise sparsity levels & Weight Histograms

In this section, we examine the layer-wise sparsity observed for ResNet-50 on CIFAR-10 across the following pruning rates: 99.75%, 99.01%, and 98.13%. As illustrated in Figure 13, the overall sparsity hierarchy is maintained, displaying a decreasing trend in sparsity from the initial layers down to the final layer, where this pattern is interrupted. We hypothesize that earlier layers retain more weights due to their critical role in feature extraction, while deeper layers can sustain higher levels of pruning without significantly impacting overall performance. Notably, the penultimate layer experiences the highest degree of pruning, which means that it contains higher redundancy or less critical weights for performance. Furthermore, by

analyzing the weight histograms for ResNet-50 with sparsity levels of 99.01% and 99.74% in Figure 11, we observe the influence of sparsity on the weight distributions. High sparsity levels significantly alter weight distributions, demonstrating that extreme pruning not only reduces the number of active weights but also changes the underlying weight dynamics within the network.

The histograms in Figure 12 illustrates the differences in weight distributions between the pruning and regrowth stages on ImageNet with ResNet-50 at approximately 4.23% remaining weights. In the pruning stage, weights are more evenly distributed across the range of $[-0.4, 0.4]$, with a noticeable dip near zero, reflecting the removal of low-magnitude weights. In contrast, during regrowth stage the weight distribution shifts significantly, showing a sharp clustering of weights around zero, indicating the reactivation of low-magnitude weights during this process. This change in distribution correlates with a notable performance gap: the regrowth stage achieves 72.4% accuracy, while the pruning stage reaches only 66.13%, we consider the cause of this to be the fact that during the pruning process the small magnitude weights are pruned and during the regrowth phase we recover from these weights the ones that improve performance the most.

### C.2. Weight flips & Implicit regrowth

Implicit regrowth serves as the main source of noise in our network, promoting diverse topologies throughout the training process. In Figure 10, we identify patterns in flip frequency, such as the lower number of flips at the start of training. This behavior is anticipated, as pruning a critical weight early on allows its features to be more readily absorbed by other weights. Around iteration 14, we notice a plateau followed by a brief decline in weight flips, which we attribute to the network stabilizing during this phase.

As training progresses and the number of parameters declines, the per-weight flip frequency continues to increase, while the overall flip frequency remains relatively steady, resulting in a continue increase of the per-weight flip frequency. The regrowth phase is marked by a sharp decrease in the total number of flips as the network stabilizes and the learning rate of flow parameters diminishes toward zero. This pattern is visible between iterations 70 and 130, alongside a gradual increase in the number of parameters.

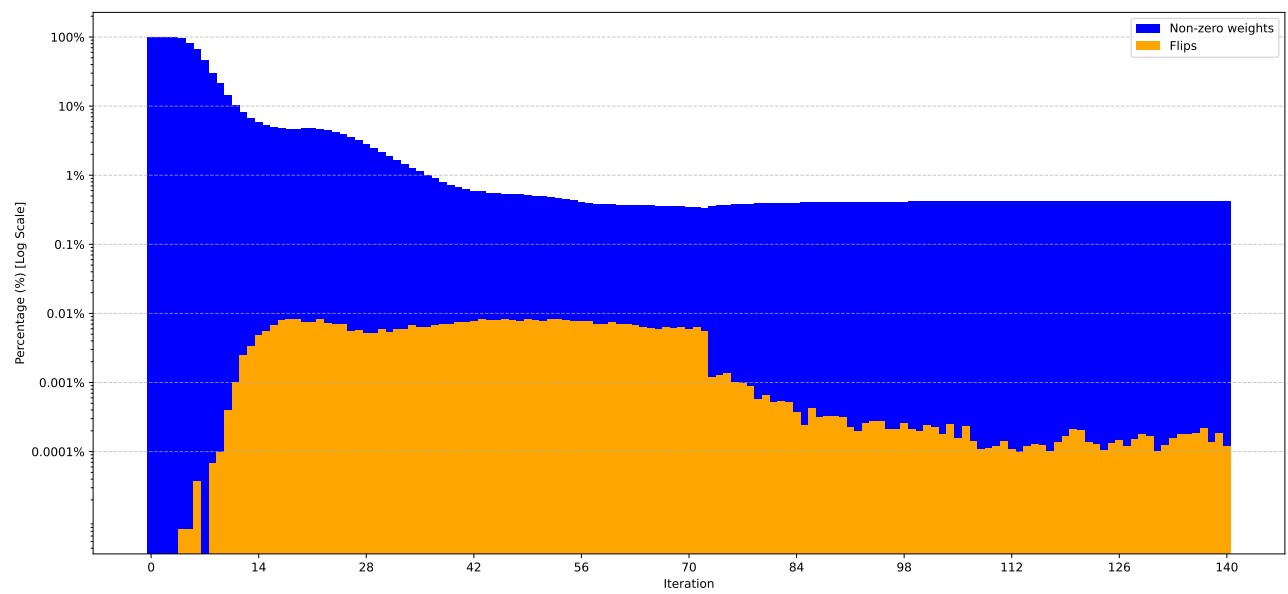

*Figure 10.* Frequency of Flips: The blue histogram represents the percentage of remaining parameters on a logarithmic scale, while the orange histogram illustrates the ratio of parameter flips per iteration relative to the total number of network parameters, also on a logarithmic scale. In our figure, one iteration is equivalent to the aggregation of 100 actual training iterations. We aggregate iterations to present the flip data in a more manageable way.

In Figure 10 we can observe the behavior of *flow* in relation to the gradients of $t$ values. Two specific type of weights emerge, as we stated in the methodology Section 3.2. Note that negative values of the gradients translate into positive updates for $t$ values and vice-versa. The first type of weight can be seen in the top-left and bottom-right diagrams in Figure 14, where

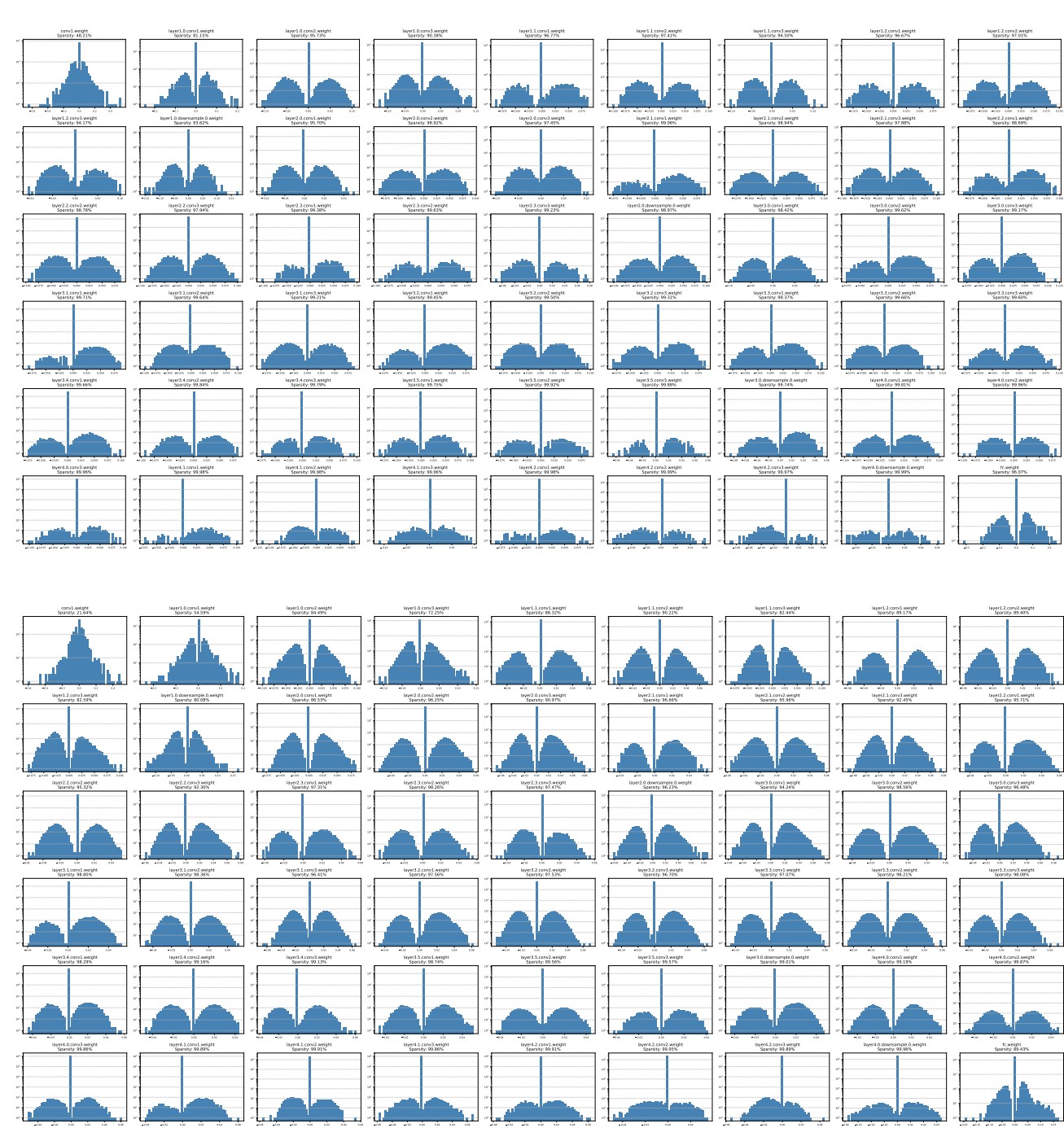

*Figure 11.* Weight values histograms of ResNet-50 on CIFAR-10 at Different Sparsity Levels. Top 99.75% sparsity, bottom 99.1% sparsity. We can observe a reshape of weight distributions

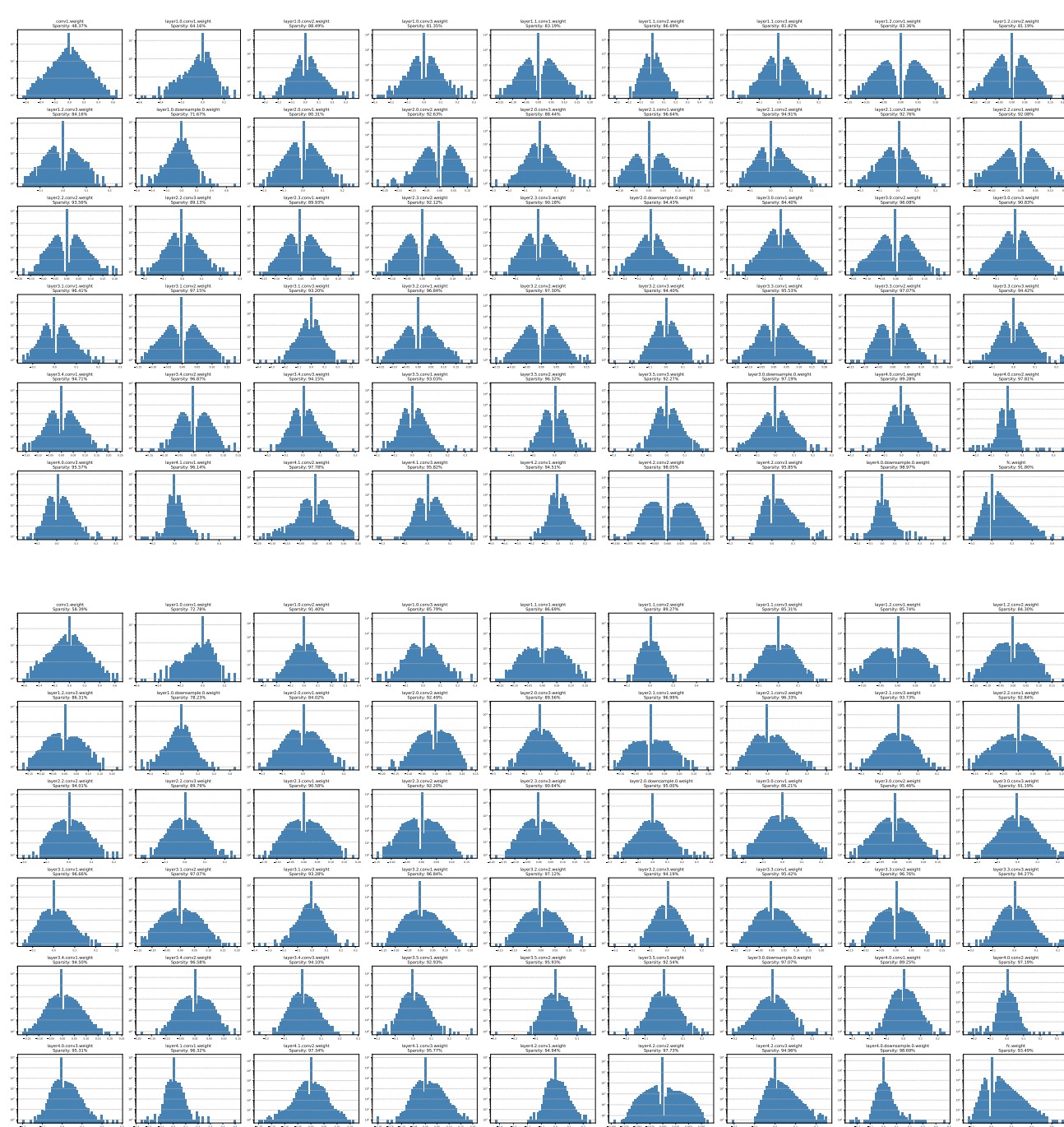

*Figure 12.* Weight Histograms: The upper figure depicts ResNet-50 during the pruning phase, achieving an accuracy of 66.13%. In contrast, the lower figure shows ResNet-50 in the regrowth phase, attaining an accuracy of 70.51%. Both phases maintain approximately 99.56% sparsity on ImageNet.

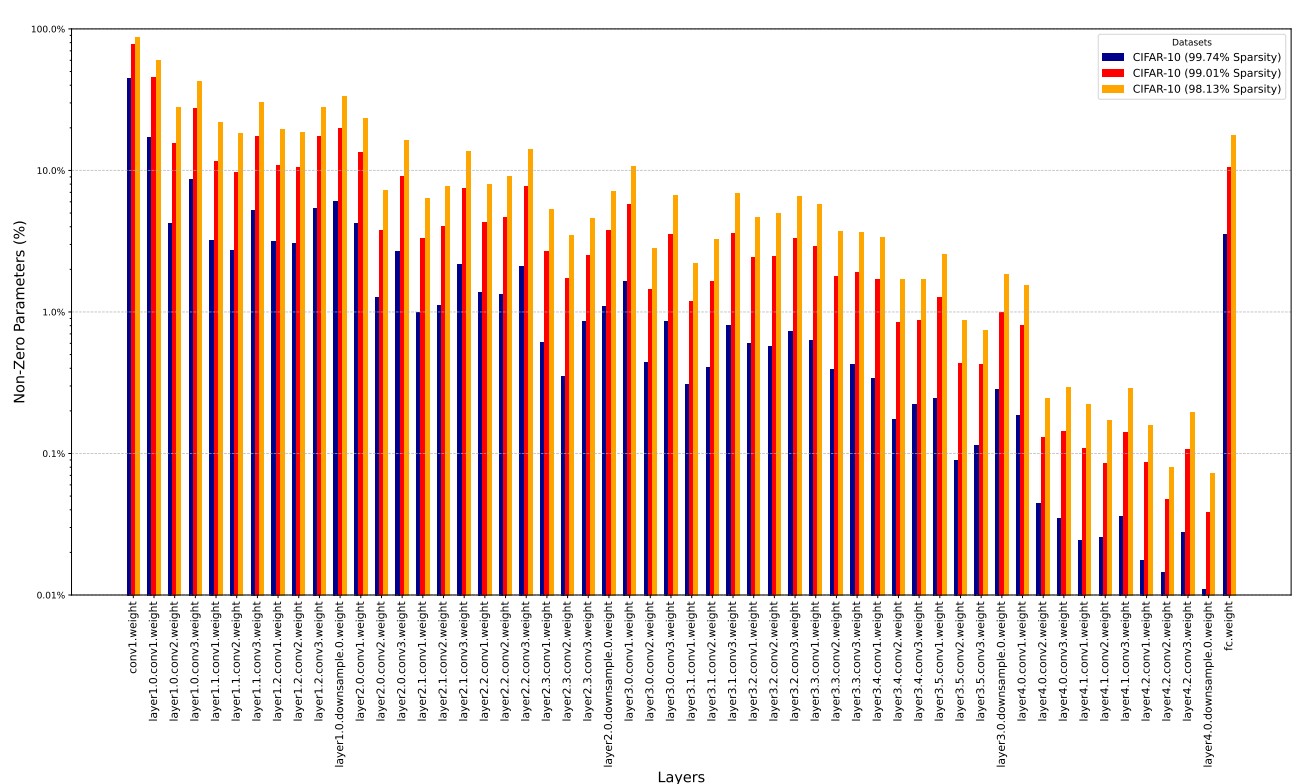

*Figure 13.* Per-layer sparsity for ResNet-50 CIFAR-10. We present 3 levels of sparsity: $99.74\%, 99.01\%$ and $98.13\%$.

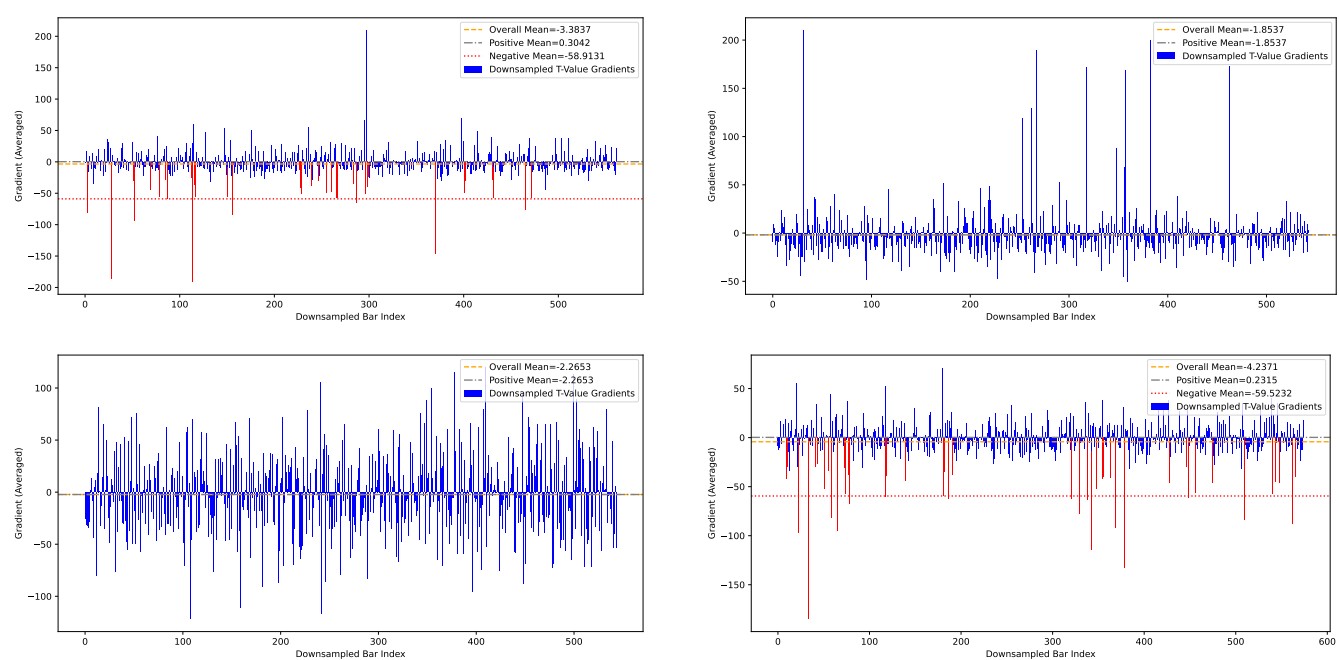

*Figure 14.* Gradient values over time corresponding to four remaining weight of a pruned network. The blue values represent gradients while $t_i > 0$, while red values represent gradients for $t_i \leq 0$. We can observe that red gradients, if they exist for that weight, have an average with very high magnitude, which is the flow $\mathcal{F}(\theta_i, \mathcal{T})$, while positive gradients $\mathcal{M}(\theta_i, \mathcal{T})$ are much smaller, but in some cases big enough to oppose pressure for several iterations.

the gradient $\mathcal{M}(\theta_i, \mathcal{T})$ does not oppose significant pressure for $t_i > 0$. This leads to the weight being pruned multiple times, which coincides, with large negative values in the gradient, which push $t_i$ back over 0. The second type of weight, as common as the first one, does not get pruned at all. In this case, $\mathcal{M}(\theta_i, \mathcal{T})$ averaged over several iterations, attempts to increase the magnitude of the weight, therefore increasing $t_i$ at the same time, which leads to the weight not being pruned at all. We can see that in this case the overall magnitude of the gradients is below $-1.5$, which in our experiment was enough to resist pressure.

# D. Training setup and reproducibility

In this section, we summarize the details regarding hyperparameters, optimizers and initializers used in our experiments.

*Table 4.* Comparison of experimental settings and results across various datasets (CIFAR-10, CIFAR-100, MNIST, and ImageNet-1K) using different neural network architectures. We make our notation as follows: $^w$ represent relation to weights, $^t$ represents relation to pruning values, $_p$, $_{ps}$ and $_{pe}$ represent pruning stage, pruning start and end, $_r$, $_{rs}$ and $_{re}$ regrowth stage, regrowth stage start and end. $\eta$ is learning rate, $\mathcal{S}$ scheduler and $\mathcal{O}$ optimizer. For example $\eta_{ps}^w$ represents the learning rate of $\theta$ values and the beginning of pruning state, and $\eta_p^t$ represents the learning of $t$ values during pruning stage. For flow params, we usea constant $\eta^t$ for pruning stage, while for regrowth we employ a exponential decay given by $\lambda_r^t$. For weights, we use a cosine annealing decay from the start to the end of pruning and naother cosine decay from the start of regrowth to the end of it. We find it useful to have a discontinuity when transitioning from pruning to regrowth, as it helps with training.

| Dataset | CIFAR-10 | | CIFAR-100 | | MNIST | ImageNet-1K |
|---|---|---|---|---|---|---|
| **Network** | **ResNet-50** | **VGG19** | **ResNet-50** | **VGG19** | **LeNet-300** | **ResNet-50** |
| **Acc (%)** | $93.0 \pm 0.5$ | $94.0 \pm 0.4$ | $93.0 \pm 0.6$ | $72.0 \pm 1.2$ | $75.0 \pm 1.1$ | $72.0 \pm 1.3$ |
| **Batch size** | 128 | 128 | 128 | 128 | 128 | 1024 |
| $\eta_{ps}^w$ | 0.1 | 0.1 | 0.1 | 0.1 | 0.001 | 0.1 |
| $\eta_{pe}^w$ | 0.003 | 0.003 | 0.003 | 0.003 | 0.001 | 0.003 |
| $\eta_{rs}^w$ | 0.001 | 0.001 | 0.001 | 0.001 | 0.001 | 0.001 |
| $\eta_{re}^w$ | 0.0001 | 0.0001 | 0.0001 | 0.0001 | 0.00001 | 0.0001 |
| $\mathcal{O}^w$ | SGD | SGD | SGD | SGD | ADAM | SGD |
| $\mathcal{S}_{\mathbf{p}}^w$ | Cosine | Cosine | Cosine | Cosine | Cosine | Cosine |
| $\mathcal{S}_{\mathbf{r}}^w$ | Cosine | Cosine | Cosine | Cosine | Cosine | Cosine |
| $\eta_p^t$ | 0.001 | 0.001 | 0.001 | 0.001 | 0.001 | 0.001 |
| $\eta_{rs}^t$ | 0.01 | 0.01 | 0.01 | 0.01 | 0.001 | 0.01 |
| $\lambda_r^t$ | 0.75 | 0.75 | 0.75 | 0.75 | 0.75 | 0.75 |
| $\mathcal{S}_{\mathbf{r}}^t$ | LambdaLR | LambdaLR | LambdaLR | LambdaLR | LambdaLR | LambdaLR |
| $\mathcal{O}^t$ | ADAM | ADAM | ADAM | ADAM | ADAM | ADAM |
| **Initialization** | Kaiming | Kaiming | Kaiming | Kaiming | Xavier | Kaiming |
| **Epochs** | 160 | 160 | 160 | 160 | 60 | 120 |
| #Prune end | 100 | 100 | 100 | 100 | 30 | 90 |

