# OpenReview forum: "Hyperflows: Pruning Reveals the Importance of Weights"
_ICML.cc/2025/Conference — Submitted to ICML 2025_

### Official Review · Reviewer_uJ5x · 2025-02-27

**Overall Recommendation:** 3

**Summary:**

The paper proposes a 'prune and regrow' approach during training. The concept of hyperflows and pressure is introduced. Hyperflows behave as a sort of saliency measure for each neural network weight. Pressure is used to control the sparsity of the network. Pruning during training behavior is analyzed in order to derive scaling laws.

## update after rebuttal
I have raised my score slightly.

**Claims And Evidence:**

The claims seem sufficient.

**Essential References Not Discussed:**

N/A

**Experimental Designs Or Analyses:**

The experimental design and analyses seem sufficient.

**Methods And Evaluation Criteria:**

The methods and evaluation seem sufficient.

**Other Comments Or Suggestions:**

N/A

**Other Strengths And Weaknesses:**

There is no intuition behind hyperflows other than: Inspired by the well-known insight that the value of something is not truly known until it is lost, we introduce Hyperflows, a dynamic pruning method which determines weight importance by first removing it.

Is the idea behind this approach coming from max-flows? Can the authors better give an explanation of as to the novelty or correctness of their approach. Why should Hyperflows and pressure be the way to prune something? Why or how does it tie into scaling laws? Why should a practitioner use hyperflows as opposed to some competing approach?

**Questions For Authors:**

N/A

**Relation To Broader Scientific Literature:**

There have been previous works in prune and regrow before. Though not many.

**Theoretical Claims:**

There are no significant theoretical claims in the work.

---

> ### Author Rebuttal · Authors · 2025-03-30
>
> Thank you for the insightful questions. We would like to clarify the inspiration, novelty, and practical advantages of Hyperflows:
>
> **C:** “Is the idea behind this approach coming from max-flows?”
>
> **R:** While the notion of “flow” might remind us of max-flow formulations in network theory, our approach is not directly derived from max-flow algorithms. Instead, Hyperflows is motivated by a fundamentally different idea: rather than solving a max-flow problem, we quantify a weight’s importance by measuring the aggregated gradient “flow” when that weight is temporarily removed. This flow essentially captures how much the loss changes if that weight is removed, thereby serving as a proxy for the weight’s criticality.
>
> An analogy could nevertheless be developed in relation with the max-flows algorithm. The weights with larger flow are kept, which can be interpreted as capacity.
>
> **C:** “Can the authors better give an explanation of as to the novelty or correctness of their approach?”
>
> **R:** The novelty lies in the concepts of pressure and flow, which can be used to drive pruning decisions without direct interference. Also, we show relationships between these notions in the neural pruning laws section, hinting at ties with the fundamental structure of the neural network. As to correctness, we address it through a combination of theoretical proofs and empirical validation. Notably, in our supplementary material A.1, we prove that if pruning a weight $\theta_i$ leads to a larger loss increase than pruning another weight $\theta_j$, then the aggregated gradient (flow) associated with $\theta_i$ is larger in magnitude, which reflects that important weights are identified by our method upon pruning. We will revise the related work section in the paper to clarify all this.
>
> **C:** “Why should a practitioner use Hyperflows as opposed to some competing approach?  Why should Hyperflows and pressure be the way to prune something? Why or how does it tie into scaling laws?”
>
> **R:** We believe that Hyperflows is a simple framework. It uses the same pruning hyperparameters for all networks, working well out-of-the-box as shown in our experiments, while allowing to set a desired sparsity. On the other hand, much of the value of Hyperflows resides in the theoretical concepts it derives and how they relate to each other. As opposed to a purely empirical paper focused on results, Hyperflows builds an intuition around the idea of pruning and uses pressure not only as a mean to prune but as a way to analyze the connections between the various effects of pruning which fall under Neural Scaling Laws. For example, we observe predictable final sparsity when pressure is constant and resilience of critical connections to extreme pressure levels. In one experiment, on LeNet300 MNIST, we observed that pruning the weights between the last hidden layer and the output leads to an identity, which lead to an extremely large flow. This happens because each remaining weight is tied to a class.

---

> > ### Comment · Reviewer_uJ5x · 2025-04-05
> >
> > I think the rebuttal is appropriate, I have slightly raised my score. I think the novelty of this work is high. It would be nice to have some citations giving a more intuitive tie-in for flows (if accepted) and other "Physics of AI" approaches if this has not been done already in the draft.

---

> > > ### Author Response · Authors · 2025-04-07
> > >
> > > Thank you for your positive feedback and for raising your score. We appreciate your recognition of the work’s novelty, as well as your suggestion regarding additional citations to tie flows into broader “Physics of AI” approaches, which we will address in the final version.

---

### Official Review · Reviewer_RnFA · 2025-03-11

**Overall Recommendation:** 2

**Summary:**

This work propose a novel algorithm that measuring importance of weights through observing their gradient during dynamic prunning process. The weights that are believed to be important will regrow at later stage. Overall the proposed algorithm show better performance than those compared with in this work.

## update after rebuttal
I believe the current state of the paper is not ready for publication, as the baseline used in the performance comparison is not appropriate. More broadly, I think we should avoid continually proposing new variations of pruning algorithms that yield similar results. That said, this is just my personal opinion and does not necessarily reflect the views of the broader community.

**Claims And Evidence:**

Overall the design of the algorithm is reasonable. However, the author might want to clearly compare the actual computing cost associated with this new algorithm. Is the computing cost similar to other algorithms. In addition, there are many other algorithms (Please see link below) can achieve better performance than the results shown in this paper, it should be justified why those are not inculded in this work.

https://paperswithcode.com/sota/network-pruning-on-imagenet-resnet-50-90

**Essential References Not Discussed:**

See link below.
https://paperswithcode.com/sota/network-pruning-on-imagenet-resnet-50-90
Also see:
@article{li2024pushing,
  title={Pushing the Limits of Sparsity: A Bag of Tricks for Extreme Pruning},
  author={Li, Andy and Durrant, Aiden and Markovic, Milan and Yin, Lu and Leontidis, Georgios},
  journal={arXiv preprint arXiv:2411.13545},
  year={2024}
}

**Experimental Designs Or Analyses:**

The experimental designs are quite standard. Though the authors might want to claify on the computing complexity of the algorithm.

**Methods And Evaluation Criteria:**

The method is overall reasonable.

**Other Comments Or Suggestions:**

Neural network pruning has a long history, with likely hundreds of algorithms proposed. However, given the distributed nature of neural network representations, the importance of pruning any specific weight may be questionable. Does it still make sense to invest time in developing new variations of pruning algorithms?

**Other Strengths And Weaknesses:**

N.A.

**Questions For Authors:**

I don't believe making connection to the scaling laws add value to this work.
L131: "by capturing the features lost from the permanently pruned weights, leading to larger flows" not sure what you meant by capturing the features"

**Relation To Broader Scientific Literature:**

N.A.

**Theoretical Claims:**

N.A.

---

> ### Author Rebuttal · Authors · 2025-03-30
>
> We thank the reviewer for the analysis and highlighting potential issues of the manuscript. We address them below.
>
> **C:** “The author might want to clearly compare the actual computing cost associated with this new algorithm.”
>
> **R:** We compared Hyperflows with methods that do not require additional parameters and acknowledged the downside that Hyperflows would require increased FLOPs at training due to learning more parameters. We appreciate the reviewer’s suggestion regarding the analysis of computational efficiency. In response, we conducted a detailed computational analysis, set to be included in the final paper, with the results summarized in Table 1. As shown, Hyperflows consistently utilizes a higher percentage of FLOPs during training, whereas it requires significantly fewer FLOPs at inference. This contrast is due to Hyperflows’ layer-wise sparsity distribution, which prunes more aggressively in computationally expensive layers, such as the 3×3 convolutions in bottleneck blocks. This trend is also illustrated in Figure 12 of C.2. To estimate the number of FLOPs, we approximate the backward pass as 2·fs + fd, where fs refers to the number of FLOPs associated with sparse weight tensors and fd refers to the FLOPs associated with dense t value tensors. The term 2·fs is used to approximate the backward FLOPs for sparse weights based on their forward FLOPs, following a common convention in the literature, while fd accounts for the backpropagation of the t values, which are maintained in dense form to enable potential weight regrowth. This yields a total training cost of 3·fs + fd FLOPs, indicating that Hyperflows requires at least one-third of the FLOPs compared to the dense baseline. This can be observed in the expression (3·fs + fd) / (3·fd) = fs/fd + 1/3. 3 fd is the total compute cost of a dense network, 1 fd for forward + 2 fd for backward.
>
> ## Table1
> |Method|Top-1Acc(%)|Params|Sparsity(%)|FLOPs(Test)|FLOPs(Train)|
> |---|---|---|---|---|---|
> |ResNet-50|77.01|25.6M|0.00|1.00x|1.00x|
> |GMP|73.91|2.56M|90.00|0.10x|0.51x|
> |DNW|74.00|2.56M|90.00|0.10x|-|
> |RigL|73.00|2.56M|90.00|0.24x|0.25x|
> |GraNet|74.50|2.56M|90.00|0.16x|0.23x|
> |STR|74.31|2.49M|90.23|0.08x|-|
> |**Hyperflows**|**74.90**|**2.54M**|**90.11**|**0.15x**|**0.60x**|
> |GMP|70.59|1.28M|95.00|0.05x|-|
> |DNW|68.30|1.28M|95.00|0.05x|-|
> |GraNet|72.30|1.28M|95.00|0.12x|0.17x|
> |RigL|70.00|1.28M|95.00|0.12x|0.12x|
> |STR|70.40|1.27M|95.03|0.04x|-|
> |**Hyperflows**|**72.20**|**1.13M**|**95.58**|**0.08x**|**0.52x**|
> |RigL|67.20|0.90M|96.50|0.11x|0.11x|
> |STR|67.22|0.88M|96.53|0.03x|-|
> |GraNet|70.50|0.90M|96.50|0.09x|0.15x|
> |**Hyperflows**|**70.40**|**0.92M**|**96.42**|**0.06x**|**0.49x**|
>
> x = fraction of baseline value
>
> **C:** Uncited reference A bag of tricks...
>
> **R:** We decided not to reference the paper, given the fact that their weight sharing scheme affects their definition of sparsity and therefore does not match the scope of our manuscript anymore.
>
> **C:** “there are many other algorithms which can achieve better performance”
>
> **R:** We wanted to compare with state of the art methods which use the same benchmarks as us, due to the fact that the process of adding new network architectures to the benchmarks is time consuming. Thus we selected methods that were in the computational bounds of what we could afford. Nevertheless, we agree with the reviewer and will compare Hyperflows with the methods mentioned above on ImageNet benchmark, in the final version.
>
> **C:** “Does it still make sense to invest time in developing new variations of pruning algorithms?”
>
> **R:** Pruning is driving superposition and distributed representations, which can help especially in explainable AI. Thus, we believe that sufficiently advanced network pruning and compression techniques can offer insights into how a neural network works, potentially leading to more efficient models and explainability. For example Anthropic used sparse autoencoders, which does compression, to explain LLM facts. ( https://www.anthropic.com/research#interpretability ).
>
> **C:** “... connection to the scaling laws add value to this work”, “by capturing the features lost from the permanently pruned weights, leading to larger flows”
>
> **R:** We believe that scaling laws reveal indirectly how the network compresses information as weights are pruned, leading to our analysis in the neural pruning laws section. For example, compression is, perhaps surprisingly, affected by the learning rate on the weights, even though we use a separate learning rate for t values.
> When weights are pruned, in order to minimize the loss function, the network will aim to compress the information in the remaining weights, to extract as many features as before more efficiently. Since more features are encoded in fewer weights, removing any weight, will lead to larger drops in accuracy, thus leading to a larger gradient aiming to regrow the weight (or in other words, the weight has larger flow). This phenomenon is analyzed in A1.

---

### Official Review · Reviewer_M1AH · 2025-03-12

**Overall Recommendation:** 2

**Summary:**

The authors propose Hyperflows, a pruning-during-training method. It assigns each parameter a learnable parameter to determine if a certain parameter should be pruned. The effectiveness of Hyperflows is tested across multiple datasets, including CIFAR10, CIFAR100, and ImageNet. It outperforms baseline methods in most scenarios.

**Claims And Evidence:**

Yes the claims made in the submission is supported by clear evidence.

**Essential References Not Discussed:**

No.

**Experimental Designs Or Analyses:**

I have checked the soundness of the experimental design of comparing with baselines.

**Methods And Evaluation Criteria:**

In fact, I am not entirely sure if the comparison is fair. All of the baseline methods do not require training additional learnable parameters. It requires much more computational costs compared to baselines. To provide a more comprehensive evaluation, the authors should include more dense-to-sparse methods like l-0 regularization [1], or other dense-to-sparse methods which learn masks through straight-through-estimators.

[1] Louizos, Christos, Max Welling, and Diederik P. Kingma. "Learning sparse neural networks through $ L_0 $ regularization." arXiv preprint arXiv:1712.01312 (2017).

**Other Comments Or Suggestions:**

I suggest the authors to reorganize the manuscript to make it more coherent and easier to understand.

**Other Strengths And Weaknesses:**

Strengths:
1. The overall performance of the proposed method is promising, surpassing baseline methods.

Weaknesses:
1. The organization of the paper makes it very difficult for readers to understand key details. For example:
    1. The algorithm is not shown in the main manuscript, making authors hard to understand how the algorithm is implemented and how pruning happens. Moreover, the role of values defined in Section 3.2 is unclear. If mask is solely dependent on $H(t_i) $, then how does the weight flow defined in section 3.2 contribute to the pruning process? Also, I do not see how these $\mathcal{F}$ and $\mathcal{M}$ are used during pruning or analysis.
    2. In section 3.2, the authors mention multiple topologies, how are they implemented?
    3. $\mathcal{I}$ is not defined in Section A.2.
    4. It is very confusing in Section 3.2 Equation (2) that we use gradient of $t_i$ while $H(\cdot)$ is not differentiable. The authors should move their explanations of using STE from line 176-182 to here.
2. As I mentioned in "Methods And Evaluation Criteria". To provide a more comprehensive evaluation, the authors should include more dense-to-sparse methods like l-0 regularization, or other dense-to-sparse methods which learn masks through straight-through-estimators. Or clarify why they are not needed.

**Questions For Authors:**

Can the authors explain the core difference between HyperFlow and those dense-to-sparse pruning methods that use learnable masks with STE?

**Relation To Broader Scientific Literature:**

The researchers may be interested in a novel pruning-during-training method.

**Theoretical Claims:**

I have checked the section A.2. However, some of the notations are not explained, for example, $\mathcal{I}$ is not defined anywhere.

---

> ### Author Rebuttal · Authors · 2025-03-30
>
> We thank the reviewer for the constructive feedback. We address the concerns below:
>
> **C:** "differences between Hyperflows and Learnable masks with STE dense-to-sparse methods."
>
> **R:**
>
> **Common aspects:**
> - Learnable Masks, L0 global pressure, STE for mask parameters.
>
> **Technical differences:**
> - Hyperflows uses the pressure scaler $\gamma$, not as a fixed regularization but as a network parameter that can be adjusted to control the network behavior. This allows for fine pruning control, making the network able to follow any desired pruning curve (including multiple stages of regrowth, aggressive pruning, lenient pruning, etc.). Additionally, relationships between pruning and other network metrics can be observed through the pressure, which result in Neural Pruning Laws (Section 3.3). These relationships can significantly improve the interpretability of the pruning process, which we consider valuable.
> - The Regrowth stage distinguishes Hyperflows from other STE methods that use learnable masks. This stage significantly boosts accuracy and is enabled by our scheduler, which precisely controls the pressure scaler.
> - One aspect that was of great importance to us was the explainability and theoretical buildup of Hyperflows pruning. Most existing methods, lack explanations and granular analysis on the underlying mechanisms of pruning beyond simple intuitions and ad-hoc heuristics with the sole purpose of increasing final accuracy. We aimed to tie together both the intuitive grounding (“you don’t know the value of something until you lose it”) with the concrete mechanism used in the method, i.e. aggregating gradients in the absence of the weight to reflect the performance impact of that weight. This gradient mechanism can be developed further to analyze other weight properties such as sign flipping, which we think is valuable for further research.
> - Some aspects in Hyperflows, such as gradient aggregation when the weight is pruned, might occur “under the hood” in certain L0 methods, but they are not a design choice nor exploited to their full potential.
>
> **C:** “the authors should include more dense-to-sparse methods like L0 regularization…”
>
> **R:** We agree with the reviewer that the comparison could use a separation between learnable mask or L0 methods and other methods not using additional parameters, with each one being compared independently with Hyperflows in terms of accuracy as well as computation. Our intention was for Hyperflows to be both theoretically grounded and to yield results comparable to popular state-of-the-art methods evaluated on the same benchmarks. Many of the L0 regularization methods we found had poor results or were not evaluated on the benchmarks we ran. Despite this, we do think that the paper would benefit from a broader comparison with L0 and learnable mask methods. Thus, we will compare Hyperflows with additional methods that use learnable masks and make a distinction in the tables between methods that use (or do not use) additional parameters. Our experiments confirm Hyperflows' additional training costs. However, the significantly lower inference cost may offset these expenses. See Table 1 posted for Reviewer 3 with ID: RnFA.
>
> **C:** “The algorithm is not shown.”
>
> **R:** We will add pseudocode detailing the full pruning algorithm in the main body.
>
> **C:** “If mask is solely dependent on $H(t_i)$, …how does flow contribute to the pruning process?”, “how these $F$ & $M$ are used during pruning or analysis.”
>
> **R:** $H(t_i)$ defines the binary mask in the forward pass. Our goal is to compute the gradient of $t_i$. Since $H(t_i)$ is not differentiable, to compute the gradient on $t_i$, which is denoted by $G$ in Section 3.2 (2), we need to use a STE for $H(t_i)$. Furthermore, $G$ has a different meaning when $t \le 0$ than when $t > 0$, so for clarity we denoted $G$ respectively by $F$ and $M$ in these two situations (even though they are the same gradient). The flow $F$ described in Section 3.2 is the gradient $G$ that propagates on $t_i$ when the weight is pruned and acts as an indicator of weight importance. In contrast, $M$ is the gradient $G$ of $t_i$ when the weight is not pruned, which makes $t_i$ follow the magnitude of $w_i$ (proven in A.2). We will revise section 3.2 to make this clearer.
>
> **C:** “the authors mention multiple topologies, how are they implemented?”
>
> **R:** These topologies are not explicitly implemented in the method; they are implicitly generated by the noise produced by pruning the weights, and in Section 3.2 they are offered as an explanation for how we are able to handle the interdependencies among weights.
>
> **C:** “$I$ is not defined, Section A.2.”
>
> **R:** $I$ was meant to be the output of the previous neuron; we will clarify this.
>
> **C:** “It is very confusing in Section 3.2 Equation (2) that we use gradient…”
>
> **R:** We will move the definitions and explanations of the STE in Section 3.2 to immediately follow Equation (2), where $G$ is defined.

---

> > ### Comment · Reviewer_M1AH · 2025-04-04
> >
> > Thank you for your detailed rebuttal. However, considering the current organization of the paper, the comparison analysis, and the marginal improvements demonstrated over existing baselines, I remain hesitant to increase my rating. Therefore, I will maintain my current rating.

---

> > > ### Author Response · Authors · 2025-04-07
> > >
> > > Thank you for your feedback. We performed additional experiments on ImageNet under the same conditions (pretrained weights, 100 epochs), comparing against GMP and GraNet, the strongest baselines in our training setup, and found that Hyperflows remains competitive. The updated results are shown in Table 2, provided in the rebuttal comment for Reviewer 1 (ID: vscf). Despite the numerically relatively small improvements, we believe Hyperflows is a strong competitor, with the extra advantage of being supported by solid theoretical grounding. We appreciate your review and will enhance the organization and clarity of our analysis.

---

### Official Review · Reviewer_vscf · 2025-03-13

**Overall Recommendation:** 3

**Summary:**

This work proposes a novel method for the pruning of parameters from deep neural network models. It focuses upon the principle of defining a network topology based upon each parameter having a measure which captures a tradeoff between a pruning 'pressure' which is applied to every node, as well as a measure of the 'flow' of each parameter. This 'flow' captures a gradient signal proportional to how much the loss would be affected if a parameter is removed. The behaviour of this novel method, 'Hyperflows', is examined across a range of hyperparameters, and in multiple neural network models. It is also compared against existing state of the art methods and shown to outperform many current competing alternatives.

**Claims And Evidence:**

Should the comparisons all be made fairly, all claims would be well supported by clear and convincing evidence.

Having said that, one potentially problematic concern is that the experimental methods for this work are not very clear. Therefore it is difficult to ascertain if the comparisons are fair. There is inconsistency, for example the methods mention that 160 epochs of training are used, however according to Appendix D the Imagenet trained models have a different length of training. Furthermore, the paper from which comparison results are taken for Tables 1 and 2 (Liu et al. 2022) appear to describe that they used a custom training setup (by taking trained 90 and 95% sparse networks and further tuning for 30 epochs to achieve extreme sparsenesses) which is not described here for this method. This could have significant implications for whether this method truly reaches a state of the art.

**Essential References Not Discussed:**

None which I am aware of.

**Experimental Designs Or Analyses:**

As mentioned in the claims section above, I have concerns regarding the experimental design and whether the experimental design is comparable between those models run for this paper (GMP, GraNet, Hyperflows) and the rest of the comparisons (e.g. RigL, STA etc). It appears that a number of results were taken from other papers but the experimental design is unclear and does not appear to conform to the same setup. Pruning, as pointed out by one of the references by Gale et al. 2019, can be highly dependent upon the parameterization used for the training.

**Methods And Evaluation Criteria:**

Methods and evaluation criteria are in line with existing work and appropriate.

**Other Comments Or Suggestions:**

- Figure 1 is never referenced in the text (as far as I can tell). Please do refer to it somewhere.
- The reference to Gale et al. 2019 is incorrect and shows up incorrectly in the main text.
- GMP is referenced to Gale et al. 2019 but originated in Zhu et al. 2019. May be good to reference the original work. Also to define some of these acronyms which are never made clear.

**Other Strengths And Weaknesses:**

This paper is, on the whole, written well and contributes a neat setup to the problem of finding a suitable topology for a network. Its results are also impressive, and clearly require a great deal of effort to produce. Some of the methods section is a little loose and could be more rigorous and clear.

**Questions For Authors:**

No specific questions, I see this as a neat paper but would like a great deal more clarity on how precisely the training was done for these models and therefore whether it is truly comparable to the other models in Tables 1 and 2.

**Relation To Broader Scientific Literature:**

It appears that this work is well related and embedded in the broader literature. It also appears that this work contributes to the overall field very well.

**Theoretical Claims:**

The theoretical and methodological descriptions are sufficiently correct, but are somewhat out of order. Most importantly, a derivative with respect to the heaviside function (H) is given as 1.0, before it is ever outlined that there is an assumption that a straight-through estimator is applied. The claim beforehand appears to suggest that the flow is measured based upon an exact derivative when this is in fact approximate. This could be clearer.

---

> ### Author Rebuttal · Authors · 2025-03-30
>
> We thank the reviewer for their detailed feedback and for recognizing the potential and novelty of our proposed Hyperflows method. Below we address the main concerns raised:
>
> **C:** “There is inconsistency, for example the methods mention that 160 epochs of training are used, however according to Appendix D the ImageNet trained models have a different length of training”
>
> **R:** In the main body of our experimental section, we omitted to specify that we train for 160 epochs on everything but ImageNet (we will clarify in the final version if eventually accepted). For ImageNet, we pruned the network for 90 epochs, with a further 30 epochs for regrowth, adding up to a total of 120 epochs. Note that GraNet and RigL train ImageNet for 100 epochs, which gives a slight advantage to Hyperflows. In Table 1 posted for Reviewer 3(ID: RnFA) are the newly computed results for ImageNet, where pruning takes place in the first 70 epochs, with an additional 30 epochs for regrowth. We also added the FLOPs comparison between methods in the same table.
>
> **C:** „Furthermore, the paper from which comparison results are taken for Tables 1 and 2 (Liu et al. 2022) appear to describe that they used a custom training setup (by taking trained 90 and 95% sparse networks and further tuning for 30 epochs to achieve extreme sparsenesses) which is not described here for this method”.
>
> **R:** Our comparison focused on two kinds of methods, during-training and one-shot methods.  For during-training methods, we aimed to evaluate them in the same conditions as Hyperflows, by running them in a post-training setup. We did this for GraNet and GMP but (i) the computational costs were high, since we needed to do a learning rate search as well as test both dense-to-sparse and sparse-to-sparse setup and report the best results. Furthermore, (ii) in almost all comparisons done between GraNet, GMP and the other methods, the latter underperformed, so we believe they would do so in the post-training setup as well, without changing the overall hierarchy of results presented in Table 1. For reasons (ii) and (i), we took the results from GraNet [2106.10404], that trained the networks under the same number of epochs, but not in a post-training setup. Nevertheless, we agree with the reviewer about the need of a rigorous design of experiments and will rectify the results, by running all the methods in the final version of the paper, for both Table 1 and Table 2.
> Furthermore, for one-shot pruning, we considered that our post-training setup is not suitable, but we decided to report their original results since many during-training pruning methods also compare to one-shot methods. If the reviewer considers that one-shot pruning methods results are of no interest, we could remove them from the final paper.
>
> **C:** “The claim beforehand appears to suggest that the flow is measured based upon an exact derivative when this is in fact approximate. This could be clearer.”
>
> **R:** To bring more clarity, we will move in the final version the definitions and explanations of the STE from Section 3.2 just after (2), where G is defined.
>
> **C:** “The reference to Gale et al 2019 is incorrect”, “GMP is referenced to Gale et al 2019”,“Figure 1 is never referenced in the text”
>
> **R:** Thank you for noticing, we will rectify and make sure to reference the original papers. Moreover, the reference is done just under the figure, on lines 203-204, to make it easier to find we will move the figure after the reference.

---

> > ### Comment · Reviewer_vscf · 2025-04-02
> >
> > Thank you for your detailed responses. Given that the comparisons in the submitted version were indeed apples to orange (i.e. quite different in retraining setup) and the new Table of performances demonstrates that GraNet can often outperform Hyperflows, I maintain my current recommendation and do not upgrade it. This is an interesting approach but I am hesitant to suggest an outright acceptance.
> >
> > I would be in favour of an overhaul of the results so that a clear apples to apples comparison could be done. This would ideally show the different methods with precisely the same training recipe applied. I am aware that this can be extremely computationally intensive but in the field of pruning comparisons this clarity is absolutely necessary.

---

> > > ### Author Response · Authors · 2025-04-07
> > >
> > > Thank you for your insightful feedback. First, we would like to clarify that the 30-epoch fine-tuning setup for 90% and 95% sparse networks mentioned by the reviewer was employed solely to assess plasticity and did not form part of GraNet’s main pruning pipeline, which starts from either a 50% randomly initialized sparse network or a fully dense network. In other words, there are two main distinctions between our initial training setup and GraNet’s: first, Hyperflows used pretrained weights, and second, our method was trained for 120 rather than 100 epochs on ImageNet.
> > >
> > > To address these concerns, we conducted additional ImageNet experiments following the Hyperflows setup (pretrained weights, 100 epochs), as summarized in Table 2. In these experiments, GraNet and GMP show only slight gains from pretraining, with Hyperflows outperforming GMP and remaining close to GraNet. These methods were chosen because they were the strongest competitors, and we do not expect the remaining experiments to alter the current performance hierarchy.
> > >
> > > In the final version of the manuscript, we will run the remaining experiments under identical conditions to ensure a fair, apples‑to‑apples comparison.
> > >
> > >
> > > **Table 2**
> > >
> > > | Method      | Top-1Acc(%) | Params  | Sparsity(%) | FLOPs(Test) | FLOPs(Train) |
> > > |-------------|------------:|--------:|------------:|------------:|-------------:|
> > > | ResNet-50   | 77.01       | 25.6M   | 0.00        | 1.00x       | 1.00x        |
> > > | **GMP**     | **74.09**   | **2.56M** | **90.00**  | **0.10x**   | **0.51x**    |
> > > | DNW         | 74.00       | 2.56M   | 90.00       | 0.10x       | -            |
> > > | RigL        | 73.00       | 2.56M   | 90.00       | 0.24x       | 0.25x        |
> > > | **GraNet**  | **74.48**   | **2.56M** | **90.00**  | **0.16x**   | **0.23x**    |
> > > | STR         | 74.31       | 2.49M   | 90.23       | 0.08x       | -            |
> > > | **Hyperflows** | **74.90** | **2.54M** | **90.11** | **0.15x**   | **0.60x**    |
> > > |||||||
> > > | **GMP**     | **70.87**   | **1.28M** | **95.00**  | **0.05x**   | **-**        |
> > > | DNW         | 68.30       | 1.28M   | 95.00       | 0.05x       | -            |
> > > | **GraNet**  | **72.54**   | **1.28M** | **95.00**  | **0.12x**   | **0.17x**    |
> > > | RigL        | 70.00       | 1.28M   | 95.00       | 0.12x       | 0.12x        |
> > > | STR         | 70.40       | 1.27M   | 95.03       | 0.04x       | -            |
> > > | **Hyperflows** | **72.20** | **1.13M** | **95.58** | **0.08x**   | **0.52x**    |
> > > |||||||
> > > | RigL        | 67.20       | 0.90M   | 96.50       | 0.11x       | 0.11x        |
> > > | STR         | 67.22       | 0.88M   | 96.53       | 0.03x       | -            |
> > > | **GMP**     | **70.39**   | **0.90M** | **96.50**  | **-**   | **-**        |
> > > | **GraNet**  | **70.79**   | **0.90M** | **96.50**  | **0.09x**   | **0.15x**    |
> > > | **Hyperflows** | **70.40** | **0.92M** | **96.42** | **0.06x**   | **0.49x**    |

---

### Decision · Program_Chairs · 2025-05-01

**Decision:**

Reject

**Comment:**

Overall the reviews of the paper were borderline and mixed even post-rebuttal, with some reviewers highlighting the novelty and motivation as reasons to lean towards accept, and other reviewers not finding the methodology to be sufficiently different from existing methodology.  Reviewers were more consistent in the post-rebuttal discussion however, where most of the reviewers made it clear that they did not believe this work was appropriate for acceptance at this stage, citing missing baselines, a lack of experiments with larger models/datasets, unclear comparisons between methods with different amounts of training.